# Calcium-permeable channelrhodopsins for the photocontrol of calcium signalling

Rodrigo G. Fernandez Lahore [1] ✉, Niccolò P. Pampaloni [2,3,8], Enrico Schiewer [1,8], M.-Marcel Heim [4,8], Linda Tillert [1,5], Johannes Vierock[1,5], Johannes Oppermann [1], Jakob Walther [6], Dietmar Schmitz[5,7], David Owald [4], Andrew J. R. Plested [2,3], Benjamin R. Rost [7] & Peter Hegemann [1]

Channelrhodopsins are light-gated ion channels used to control excitability of designated cells in large networks with high spatiotemporal resolution. While ChRs selective for $H^+$, $Na^+$, $K^+$ and anions have been discovered or engineered, $Ca^{2+}$-selective ChRs have not been reported to date. Here, we analyse ChRs and mutant derivatives with regard to their $Ca^{2+}$ permeability and improve their $Ca^{2+}$ affinity by targeted mutagenesis at the central selectivity filter. The engineered channels, termed CapChR1 and CapChR2 for calcium-permeable channelrhodopsins, exhibit reduced sodium and proton conductance in connection with strongly improved $Ca^{2+}$ permeation at negative voltage and low extracellular $Ca^{2+}$ concentrations. In cultured cells and neurons, CapChR2 reliably increases intracellular $Ca^{2+}$ concentrations. Moreover, CapChR2 can robustly trigger $Ca^{2+}$ signalling in hippocampal neurons. When expressed together with genetically encoded $Ca^{2+}$ indicators in *Drosophila melanogaster* mushroom body output neurons, CapChRs mediate light-evoked $Ca^{2+}$ entry in brain explants.

Channelrhodopsins (ChRs) constitute a large group of microbial rhodopsins that exhibit light-dependent ion channel activity[1,2]. Several natural and engineered ChRs have been shown to exhibit a higher selectivity for $H^{+3}$, $Na^{+4}$, or $Cl^{-5–7}$ compared to the originally discovered ChR1 and ChR2 of *Chlamydomonas reinhardtii*. More recently, ChRs with a high $K^+$-selectivity were also discovered[8–10]. Accordingly, when expressed in neuronal cells or tissues, ChRs enable multidirectional manipulation of electrical cellular activity. Whereas non-selective cation-conducting ChRs (CCRs) generally depolarize cells and promote action potential firing, anion-conducting ChRs (ACRs) and $K^+$-selective ChRs (KCRs) suppress spiking upon illumination[8,9,11]. However, to date, no $Ca^{2+}$-selective ChRs have been identified or engineered. Although several ChR variants with low to moderate calcium conductance have been reported[12–15], they are not selective enough for $Ca^{2+}$ to be used as genetically encoded calcium actuators (GECAs) in the presence of $Na^+$. Moreover, the mechanism of calcium permeation in CCRs, and of $Ca^{2+}$ competition with other cations, has been neglected and is thus poorly understood[16,17]. The lack of $Ca^{2+}$-selective ChRs contrasts the widespread role of calcium in cellular signalling, metabolic excitation, neuronal plasticity, muscle contraction, apoptosis, and cell growth[18–20]. In optogenetics, ChR-based calcium channels would have broad applicability in the reversible and spatiotemporally defined control of $Ca^{2+}$ dynamics and associated downstream processes. Some non-channelrhodopsin GECAs have been reported, but they all suffer from drawbacks such as low calcium release[21],

[1]Institute of Biology, Experimental Biophysics, Humboldt-Universität zu Berlin, Berlin, Germany. [2]Molecular Neuroscience and Biophysics, Leibniz-Institut für Molekulare Pharmakologie, Berlin, Germany. [3]Institute of Biology, Cellular Biophysics, Humboldt-Universität zu Berlin, Berlin, Germany. [4]Institute of Neurophysiology, Charité – Universitätsmedizin Berlin, Berlin, Germany. [5]Neuroscience Research Center, Charité – Universitätsmedizin Berlin, Berlin, Germany. [6]Department of Neurology with Experimental Neurology, Charité – Universitätsmedizin Berlin, Berlin, Germany. [7]German Center for Neurodegenerative Diseases (DZNE), Berlin, Germany. [8]These authors contributed equally: Niccolò P. Pampaloni, Enrico Schiewer, M.-Marcel Heim. ✉e-mail: r.fernandezlahore@googlemail.com

crosstalk with other molecular targets[22–26] or low temporal resolution[27–30].

Our goal was to develop Calcium-permeable ChannelRhodopsins (CapChRs) that allow a light-driven increase in cytosolic calcium with temporal precision in the sub-second range. The engineered CapChRs we report display strong inward-rectifying Ca²⁺ translocation but only meagre sodium permeation, even in the low [Ca²⁺]ₑ and high [Na⁺]ₑ conditions found in many physiological model systems. We demonstrate the potential of CapChRs as optogenetic tools for the photocontrol of calcium signalling in mammalian neurons and *Drosophila melanogaster*.

## Results

### Ca²⁺-flux in cation-conducting channelrhodopsins

To identify cation-conducting ChRs that could serve as templates for further engineering, we performed Fura-2-AM calcium imaging on cultured ND7/23 cells expressing *Cr*ChR2 WT (C2) or several other CCRs (Fig. 1). We employed *Cr*ChR2 L132C (C2-LC, reported as Calcium-translocating Channelrhodopsin or CatCh[12]) as a benchmark for the light-evoked Ca²⁺ influx in the presence of high extracellular Ca²⁺ (70 mM). In line with previous studies[12,31], C2-LC exhibits enhanced Ca²⁺-permeation when compared to the parental C2 (Fig. 1C, D) and outperformed most of the other tested wild-type variants. However, both *Co*ChR[32] and *Ts*ChR[32] showed larger calcium conductance than C2

and similar to C2-LC (Fig. 1C, D). These results recommended *Ts*- and *Co*ChR as promising templates to develop a Ca²⁺-selective ChR. In contrast, most other widely used CCRs and derivatives including bReaChES, ChRmine, C1C2, CheRiff, ReaChR, ChroME and C1V1 showed little to no Ca²⁺-conductance in this assay (Fig. 1C, D)[33–39].

### Calcium permeation of previously described ChR variants

Due to abundant mutational studies and the availability of closed-state crystal structures[34,40], we first aimed at increasing Ca²⁺-permeation in C2. Several residues have been linked to an improved Ca²⁺-conductance upon mutation, including S63[13], L132[12], D156[15], N258[13] and F219[14] (Fig. 1A). Additionally, the D139H mutation in *Ps*ChR (homologous to D156H in C2) at the retinal binding pocket was reported to increase calcium permeability[15]. However, across these studies, Ca²⁺ vs. Na⁺ permeation has not been uniformly examined. Thus, to elucidate the cation-conducting properties of selected mutants, we performed whole-cell voltage-clamp experiments in four ionic conditions: 140 mM [NaCl]ₑ, 70 mM [CaCl₂]ₑ (at pH 7.2 and 9), or 140 mM [NMGCl]ₑ (NMG: N-Methyl-D-glucamine). In this context, NMG⁺ was used as a non-conducted cation to assess the permeation of H⁺ at pH 7.2. In C2-LC (Fig. S3B), photocurrents were smaller after shifting from high Na⁺ to high Ca²⁺ and the reversal potential ($E_{rev}$) remained unchanged in calcium buffer when compared to C2, suggesting that sodium is still preferred over calcium at different voltages (Figs. S3A, B,

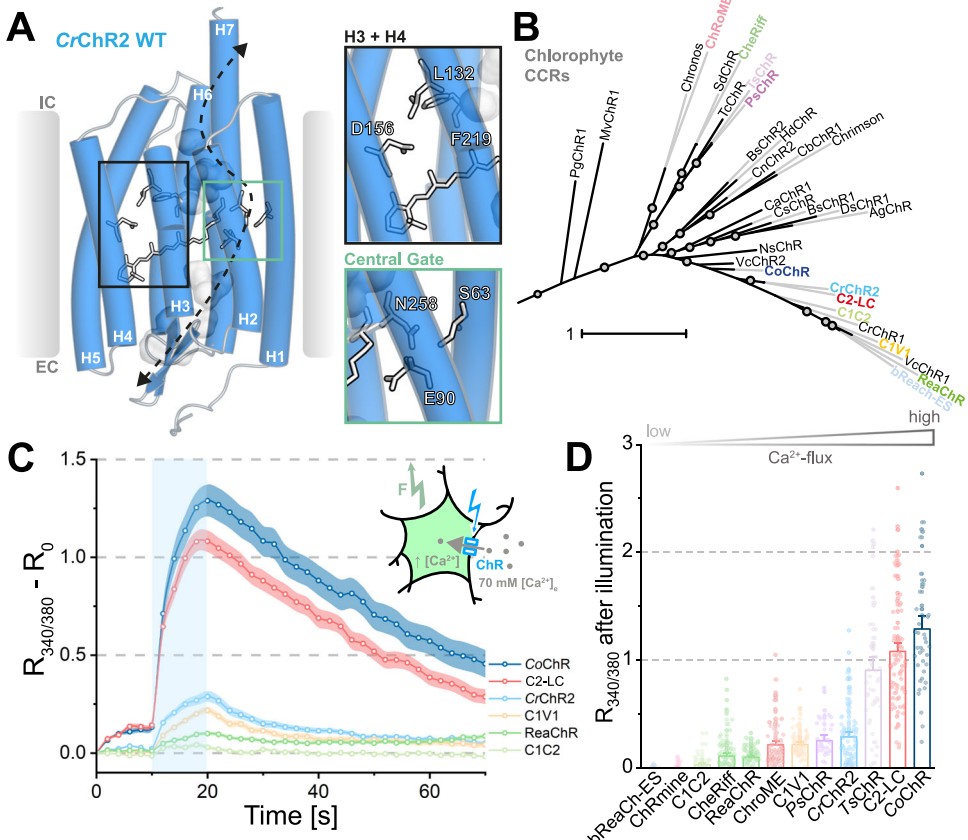

**Fig. 1 | Calcium flux at high [Ca²⁺]ₑ in selected CCRs. A** *Cr*ChR2 crystal structure (PDB ID: 6EID, https://doi.org/10.2210/pdb6EID/pdb) with proposed ion permeation pathway (black arrow) and water densities after 10 ns MD simulation (grey mesh). Amino acids in stick representation have been previously associated with calcium permeation. Insets, right: H3 and H4) Helix 3 and Helix 4 with residues L132, D156 and F219 (Helix 6) and Central Gate) Central Gate formed by S63, E90 and N258. **B** Unrooted phylogenetic tree of chlorophyte CCRs. Coloured entries represent ChRs with calcium measurements in panels (**C**) and (**D**). Overview of depicted CCRs can be found in Supplementary Fig. 1. **C** Fura-2 imaging of calcium

influx for different ChRs expressed in Fura-2-AM loaded ND7/23 cells probed with 10 s of 470 nm illumination (-0.08 mW/mm², saturating conditions for C2-LC, indicated by blue bar) at 70 mM [CaCl₂]ₑ (mean ± SEM). **D** Quantified peak responses (as in panel **C**) after 10 s of illumination. Each dot represents one cell (mean ± SEM). Number of replicates: bReaCh-ES: $n = 78$; ChRmine: $n = 37$; C1C2: $n = 45$; CheRiff: $n = 97$; ReaChR: $n = 89$; ChroME: $n = 104$; C1V1: $n = 72$; *Ps*ChR: $n = 33$; *Cr*ChR2: $n = 104$; *Ts*ChR: $n = 49$; C2-LC: $n = 91$; *Co*ChR: $n = 43$. $n = $ X biologically independent cells. EC extracellular, IC intracellular, $R_{340/380}$ Fura-2 ratio, $R_0$ Fura-2 ratio of first acquisition.

S7A, C). In agreement with a previous study[17], we thus infer that the main benefit of the C2-LC mutant for optogenetic applications is the reduced inactivation, which leads to larger stationary currents under all buffer conditions. The helix 6 mutant F219Y, for which an increased permeability ratio $P_{Ca}$ / $P_{Na}$ and accelerated channel closing was reported[14], showed only subtle differences in ion selectivity compared to C2 (Figs. S3D, S7). D156H ("XXM", C2-DH[15]), a mutation close to the retinal C7 = C8 bond, caused an increased $I_{Ca}/I_{Na}$ of ≈ 0.6 (Figs. S3C, S7) but retained a substantial Na⁺-conductivity (Fig. S3C). In contrast, the homologous mutation D139H in *Ps*ChR (Ps-DH) reduced calcium permeation and resulted in slow kinetics (Figs. S3H, J, S7).

In the second set of experiments, we studied the photocurrents of central gate mutants (Fig. 1A). The mutations N258E and N258D promoted Ca²⁺-permeation at −80 mV, with limited proton contribution and unchanged kinetics (Figs. S3E, F, S7). Whereas the close-by S63E mutation caused poor expression, the S63D (C2-SD) mutant was reliably expressed and showed improved Ca²⁺ conductance during stationary photocurrents without substantial proton permeation (Figs. S3G, S7). In addition, at negative holding voltages, the stationary Ca²⁺ permeation surpassed that of Na⁺ (Figs. S3G, S7). However, C2-SD photocurrents retained strong inactivation (like C2) at high [NaCl]ₑ and the peak current was of similar amplitude at high [CaCl₂]ₑ (Figs. S3G, S4), suggesting transient conductance of sodium and protons early during illumination of dark-adapted cells. In contrast, mutation of the homologous position S46D in *Ps*ChR did not promote calcium permeation (Figs. S3I, S7).

Based on this analysis, we confirmed that mutations at the central gate of C2 have a strong impact on the Ca²⁺ permeation but the gate likely differs between ChR variants. This idea gains support from previous work where E90 of the central gate, in close proximity to S63 and N258, was disclosed as critical for cation selectivity. Varying E90 mutations modified either the Na⁺/H⁺ ratio or even converted C2 into an anion channel[5,41]. Similarly, homotetrameric Na⁺-selective channels have been converted into Ca²⁺ channels by the introduction of negative charges to their selectivity filters[42,43] (Fig. S5). We reasoned that a related principle underlies discrimination between monovalent and divalent cations in C2, despite its functionality as a monomer with an asymmetric pore[34,40,44] (Figs. 1A, S5).

## Engineering of ChR-based calcium actuators

Based on the characterization outlined above, we combined selected mutations to improve calcium selectivity. We started from *Cr*ChR2-L132C-T159C (C2-LC-TC) due to its enhanced retinal binding, better membrane targeting and reduced desensitization[45]. Importantly, previous studies on C2-LC-TC revealed that the contribution of Ca²⁺ to the photocurrent is low or even negligible at typical vertebrate ionic conditions (120 mM [NaCl]ₑ, 2 mM [CaCl₂]ₑ and 2 mM [MgCl₂]ₑ), principally due to high Mg²⁺ affinity ($K_m$ = 2 mM)[17]. To enhance Ca²⁺ influx even at low extracellular concentrations, we included the S63D mutation (C2-SD-LC-TC). Subsequent measurements indicated an enhanced Ca²⁺ conductance and reduced desensitization for C2-SD-LC-TC (Fig. S8A, B). However, the $I_{Ca}/I_{Na}$ ratio was still below that of C2-SD (Fig. S8G). Similarly, the combination of S63D with D156H (C2-SD-DH) showed a reduced $I_{Ca}/I_{Na}$ ratio and slow channel closure kinetics (Fig. S8C, D, G, H). Based on our initial voltage-clamp experiments (Fig. S3), and following the logic outlined above, an additional negative charge in the proximity of E90 might further increase relative Ca²⁺ permeation (Fig. S5). Accordingly, we introduced a carboxylate at the N258 position in C2-SD-LC-TC (Figs. 2A, S5). The inclusion of the N258D mutation impaired expression, while the combination with N258E exhibited good expression and membrane targeting. The resulting C2-SD-LC-TC-NE mutant had enhanced inward Ca²⁺ permeation with strong inward rectification (Fig. 2D, E), similar to that of the single mutant C2-SD (Fig. S3G). It bears mentioning that the $I_{Ca}/I_{Na}$ of C2-SD-LC-TC-NE was not significantly different from that of C2-SD in

stationary photocurrents. However, the quadruple mutant exhibited two crucial advantages: reduced Na⁺ contribution to the $I_{peak}$, and elevated $I_{stat}$ current densities (Fig. S4). In comparison to C2, sodium permeation was reduced for both peak and stationary photocurrents, even though currents were generally larger (Fig. S4). Moreover, the quadruple mutant showed a more positive $E_{rev}$ in the presence of extracellular calcium (Fig. S7C), although far from the Nernst potential expected of a calcium channel under the given conditions ($E_{Nernst}$ for Ca²⁺ > +100 mV). Ergo, we inferred that C2-SD-LC-TC-NE preferentially conducts calcium at negative membrane voltages, making it a potential calcium actuator in cells with a negative resting potential.

In addition, we produced a slow channel variant by adding the D156H mutation (Fig. S8E–H). This mutant, C2-SD-LC-DH-TC-NE, retained high calcium permeability and should be useful for experiments requiring high operational light sensitivity and low temporal resolution. However, since faster kinetics are advantageous for many physiological applications due to enhanced temporal resolution, we focused subsequent work on the quadruple mutant C2-SD-LC-TC-NE.

Next, we transferred the set of mutations to *Co*ChR, due to the elevated calcium permeation of the wt protein (Fig. 1B, C). *Co*ChR displays photocurrents similar to C2, with the largest photocurrents at 140 mM [NaCl]ₑ (Fig. S2E, F) but low H⁺ contribution as seen after shifting from pHₑ 7.2 to 9 (Fig. S2F). The analogous mutation to C2-LC (*Co*ChR L112C) was reported to behave similarly, with reduced inactivation and enlarged stationary currents[46]. However, it only increased $I_{Ca}/I_{Na}$ to ≈ 0.4 (Fig. S6). Additionally, neither *Co*ChR S43D, L112C T139C nor N238E increased the $I_{Ca}/I_{Na}$ to the extent of C2-SD (Fig. S6C). Consequently, based on our initial calcium imaging results (Fig. 1), we added all four mutations to both *Co*ChR (Fig. 2A) and *Ts*ChR (*Co*ChR S43D L112C T139C N238E and *Ts*ChR S45D L114C G141C N240E, Fig. S1). Whereas the four *Ts*ChR mutations impaired expression in ND7/23 cells, the resulting *Co*ChR (*Co*-SD-LC-TC-NE) expressed well with reliable membrane localization. It exhibited improved calcium permeation at a negative voltage, similar to C2-SD-LC-TC-NE (Fig. 2D, H). Recordings on *Co*-SD-LC-TC-NE also revealed a high relative calcium permeation ($I_{Ca}/I_{Na}$ ≈ 1.9), an improved positive $E_{rev}$-shift in calcium buffer in comparison to the WT, and low proton permeability (Figs. 2J, K, S2H). According to these properties, we designated C2-SD-LC-TC-NE and *Co*-SD-LC-TC-NE as CapChR1 and CapChR2, respectively (Calcium permeable ChannelRhodopsins) and the slow CapChR1 (C2-SD-LC-DH-TC-NE) as sCapChR1. We found the permeability for K⁺ to be similarly low to that of sodium and the Mg²⁺-conductance to be poor in both CapChRs (Figs. 2F, J, S9). The modifications had little impact on the action spectra of the proteins, although CapChR2 shows a 4 nm red-shift and band narrowing at high [CaCl₂]ₑ (Fig. S10), which was not seen in the purified protein (Fig. S18).

## CapChRs exhibit inward-rectifying calcium permeation with low conductance for sodium

Following the dissection of the relative cation permeation, the particular condition of low Ca²⁺ with high Na⁺, as found in many vertebrate host systems, merited further characterization. To test ion competition in these conditions, we recorded I(E)-relationships of the CapChRs over a range of different Ca²⁺/Na⁺ ratios in comparison with C2-LC (Fig. 3A–C). For C2-LC, we observed large inward photocurrents at −80 mV in presence of 140 mM [NaCl]ₑ and a decrease in amplitude upon replacement of Na⁺ by Ca²⁺ (Fig. 3A, D, G), indicating poor conductance for Ca²⁺ relative to Na⁺. This was accompanied by a negative $E_{rev}$-shift ($\Delta E_{rev}$) upon complete sodium replacement (Fig. 3D). Conversely, both CapChRs exhibit a calcium-dependent increase of inward currents (Fig. 3B, C). At −80 mV, currents increased ≈3-fold for CapChR1 (Fig. 3E, H) and ≈5-fold for CapChR2 when Na⁺ was replaced with Ca²⁺, coupled with reduced negative shifts in $E_{rev}$ for CapChR2 (Fig. 3F, I). Moreover, CapChR2 features a sharp increase in $\tau_{off}$ from 87.3 ± 17.7 ms at 0 mM [CaCl₂]ₑ to 317.3 ± 17.7 ms at 70 mM [CaCl₂]ₑ

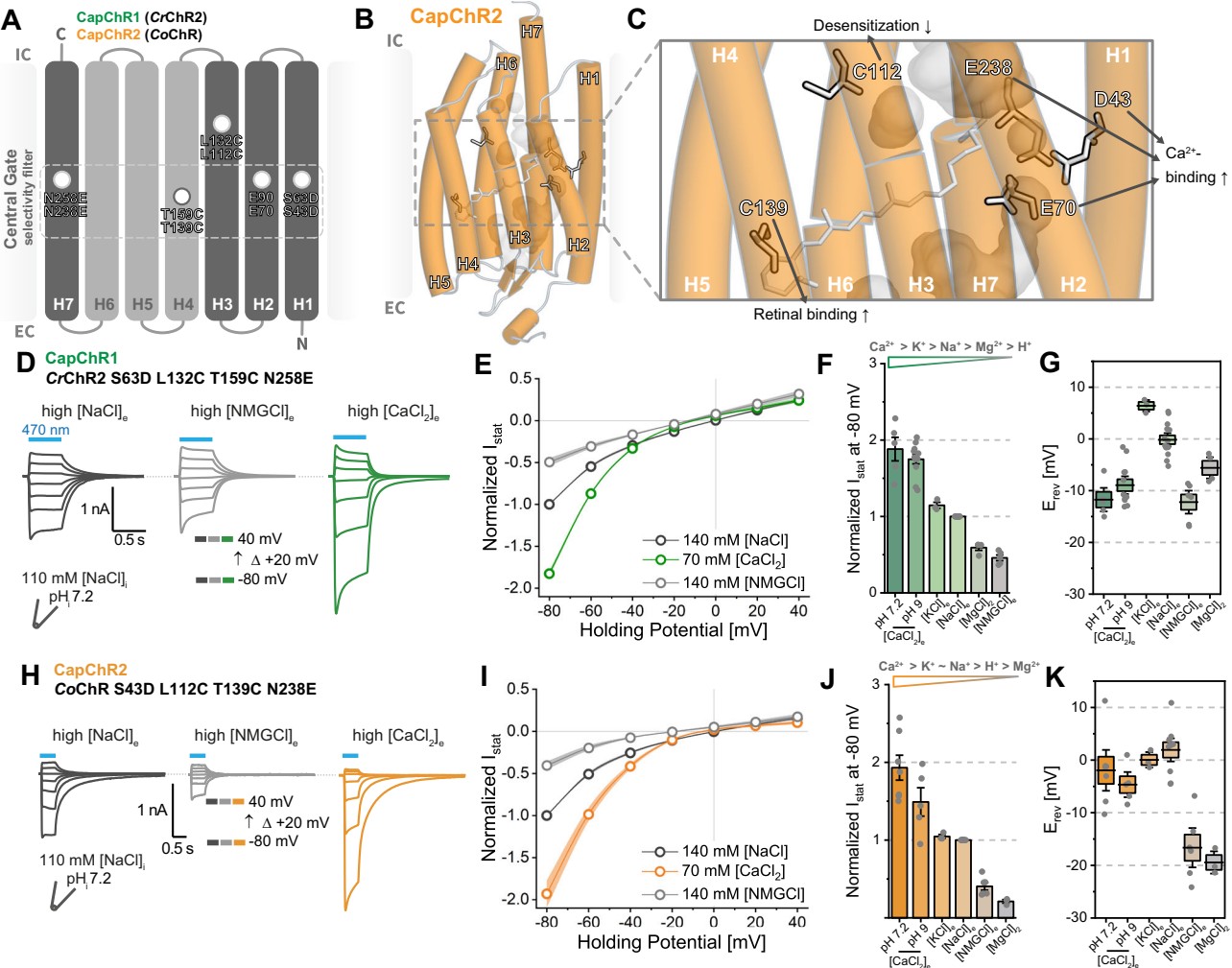

**Fig. 2 | Engineering of Calcium-permeable ChannelRhodopsins (CapChRs).**
**A** Design strategy for CapChR1 (green) and CapChR2 (orange). Pore helices coloured in dark grey and miscellaneous helices in light grey. **B** and **C** Exemplary CapChR2 homology model depicting mutated residues and their intended effects (based on structure data, PDB ID: 6EID, https://doi.org/10.2210/pdb6EID/pdb). Grey mesh represents water accessibility according to MD simulations. **D** and **H** Representative photocurrent traces of CapChR1 (green) and CapChR2 (orange) respectively, recorded from −80 to +40 mV in 20 mV steps in ND7/23 cells (blue bar: illumination with saturating, 470 nm light; -1.9 mW/mm²). **E** and **I** Current−voltage relationships for CapChR1 and CapChR2 under varying extracellular conditions (shadows represent the SEM; normalized to high [NaCl]$_e$ at −80 mV). **F** and **J** Stationary photocurrents for CapChR1 and CapChR2 at the

designated extracellular conditions (mean ± SEM). **G** and **K** Estimated reversal potential ($E_{rev}$) for CapChR1 and CapChR2 at the designated ionic conditions (Box middle line: Mean; Box outer edges ± SEM; Box whiskers: 1.5 × SEM). Number of replicates in **E** [NaCl]$_e$: $n = 14$; [CaCl$_2$]$_e$: $n = 12$; [NMGCl]$_e$: $n = 6$, in **F** [NaCl]$_e$: $n = 14$; [CaCl$_2$]$_e$ pH 7.2: $n = 12$; [CaCl$_2$]$_e$ pH 9: $n = 5$; [NMGCl]$_e$: $n = 5$; [KCl]$_e$: $n = 3$; [MgCl$_2$]$_e$: $n = 4$, in **G** ([NaCl]$_e$: $n = 14$; [CaCl$_2$]$_e$ pH 7.2: $n = 12$; [CaCl$_2$]$_e$ pH 9: $n = 5$; [NMGCl]$_e$: $n = 6$; [KCl]$_e$: $n = 3$; [MgCl$_2$]$_e$: $n = 4$), in **I** ([NaCl]$_e$: $n = 10$; [CaCl$_2$]$_e$: $n = 7$; [NMGCl]$_e$: $n = 6$), in **J** [NaCl]$_e$: $n = 10$; [CaCl$_2$]$_e$ pH 7.2: $n = 7$; [CaCl$_2$]$_e$ pH 9: $n = 5$; [NMGCl]$_e$: $n = 6$; [KCl]$_e$: $n = 3$; [MgCl$_2$]$_e$: $n = 3$, in **K**) [NaCl]$_e$: $n = 10$; [CaCl$_2$]$_e$ pH 7.2: $n = 7$; [CaCl$_2$]$_e$ pH 9: $n = 5$; [NMGCl]$_e$: $n = 6$; [KCl]$_e$: $n = 3$; [MgCl$_2$]$_e$: $n = 3$. $n = X$ biologically independent cells. EC extracellular, IC intracellular.

(Fig. 3I), perhaps caused by a high calcium affinity of the open state. Due to the extended open-state lifetime, CapChR2 provides an operational light sensitivity superior to both CapChR1 and C2-LC (Fig. S11). Moreover, we could confirm that CapChR2 lacks substantial proton permeation at pH 7.2 by measuring photocurrents in the absence of sodium and calcium (Fig. S12).

Prior studies have shown that ChRs may strongly bind divalent cations but permeate the ions only poorly (e.g. C2-LC[17]). We therefore quantified the impact of different divalent cations on the inward currents of CapChR1 and 2 (Fig. S13). In CapChR1, Ca²⁺, Mg²⁺, Ba²⁺, and Sr²⁺ had negligible impact on current amplitudes at 140 mM [NaCl]$_e$, $E_{rev}$ and channel closure kinetics (Fig. S13A–D, H). In CapChR2, current amplitudes were enhanced upon the addition of 2 mM [CaCl$_2$]$_e$ and suppressed gradually by increasing Mg²⁺, Ba²⁺, or Sr²⁺ concentrations (Fig. S13A, E, F). Ca²⁺ or Mg²⁺ retarded channel closure at −80 mV with weak effects on the $E_{rev}$ (Fig. S13G, I). Thus, we conclude that Ca²⁺ is

both bound and conducted by CapChR2, whereas Mg²⁺ and larger relatives bind and block or only weakly permeate the pore (Figs. S13, S14).

**Calcium imaging at low calcium concentrations**
To visualize the calcium permeation of CapChRs at more physiological Ca²⁺ levels (2 mM), we first compared their performance relative to CoChR and C2-LC in Fura-2-AM-loaded ND7/23 cells. In the same experiment, we additionally assessed the impact of bath Mg²⁺ on the Ca²⁺-influx (Fig. 4A, B). These measurements confirmed the superiority of CapChR2 at the resting membrane potential of cultured cells and in the presence of 2 mM Mg²⁺ (Fig. 4B, C). It is worth noting that both CapChRs exhibit strongly reduced Na⁺ permeation under similar buffer conditions (Fig. 3). In a separate approach, we imaged calcium permeation through CapChRs at 2 mM extracellular calcium in cells with membrane voltage clamped at −80 mV. Here, cells were loaded

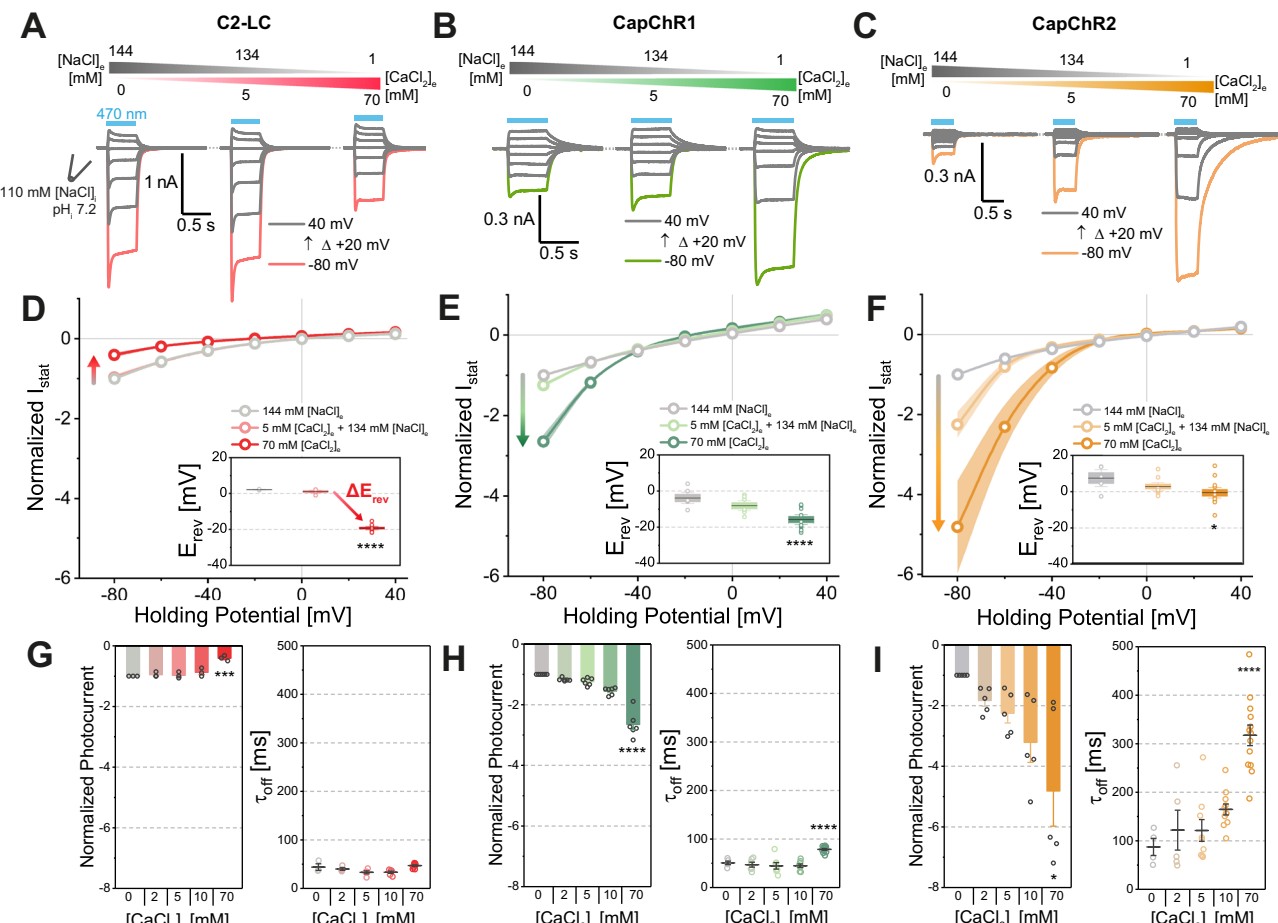

**Fig. 3 | Calcium vs. sodium permeation in C2-LC, CapChR1 and CapChR2.**
**A–C** Representative photocurrents traces of C2-LC (red), CapChR1 (green) and CapChR2 (orange), respectively, recorded from −80 to +40 mV in 20 mV steps at different calcium/sodium concentrations in ND7/23 cells (blue bar: illumination with saturating, 470 nm light; -1.9 mW/mm²). **D–F** $I(E)$ relationships with estimated reversal potentials (insets, Box middle line: Mean; Box outer edges ± SEM; Box whiskers: 1.5 × SEM) for the tested ChRs at the designated ionic conditions (mean ± SEM, normalized to currents at −80 mV and 0 mM [CaCl₂]ₑ). Dots represent mean values and shading the SEM (p values for **D**, **E** and **G**: C2-LC $p = 5.52 \times 10^{-10}$; CapChR1 $p = 4.06 \times 10^{-5}$; CapChR2 $p = 0.04$). **G–I** Left: Normalized photocurrents at −80 mV holding potential and at the denoted [CaCl₂]ₑ in the presence of sodium. Bars represent the mean ± SEM and dots represent single

measurements. Right: Channel closure kinetics at the denoted [CaCl₂]ₑ in the presence of sodium ($\tau_{off}$). Lines and bars represent the mean ± SEM and dots represent single measurements). Number of replicates for photocurrents in **D–I** C2-LC: $n = 3$ in all conditions; CapChR1: $n = 6$ in all conditions; CapChR2: $n = 5$ in all conditions, for $\tau_{off}$ in **G–I** C2-LC/CapChR1/CapChR2: 144 mM [NaCl]ₑ/0 mM [CaCl₂]ₑ: $n = 3/5/4$, 140 mM [NaCl]ₑ/2 mM [CaCl₂]ₑ: $n = 4/6/5$, 134 mM [NaCl]ₑ/5 mM [CaCl₂]ₑ: $n = 5/7/9$, 124 mM [NaCl]ₑ/10 mM [CaCl₂]ₑ: $n = 6/8/11$, 1 mM [NaCl]ₑ/70 mM [CaCl₂]ₑ: $n = 11/12/13$. p values for **G–I** (left/right): C2-LC $P = 3.33 \times 10^{-4}$; CapChR1 $P = 2.63 \times 10^{-6}/1.46 \times 10^{-6}$; CapChR2 $P = 0.01/3.93 \times 10^{-5}$. Two-sided two-group t-test (control: 144 mM [Na⁺]ₑ/0 mM [Ca²⁺]): *$P \leq 0.05$, **$P \leq 0.01$, ***$P \leq 0.001$, ****$P \leq 0.0001$. $n = X$ biologically independent cells.

with membrane-impermeant calcium indicator Fura-2 via the patch pipette and the calcium influx was monitored during illumination (Fig. 4D). At 2 mM [Ca²⁺]ₑ, both CapChR1 and CapChR2 elicited reliable calcium influx in the presence of 140 mM [Na⁺]ₑ at pH 7.2 (Fig. 4E). Even so, CapChR2-expressing cells showed ≈8-fold larger initial rise and a single 100 ms light pulse (0.008 mJ/mm² at 470 nm) almost saturated the Fura-2 response (Fig. 4E, F). Ca²⁺ permeation was larger in CapChR2-expressing cells at all tested concentrations (0.5–20 mM [Ca²⁺]ₑ), confirming stronger selective calcium permeation in the presence of high concentrations of competing Na⁺ (Fig. 4G, H, Fig. S15). In summary, these results indicate that (i) both CapChRs efficiently conduct calcium into the cell, (ii) both CapChRs could function as GECAs at 2 mM [Ca²⁺]ₑ in cells with a negative resting membrane potential and (iii) CapChR2 shows superior potential as a calcium actuator for optogenetic applications.

### Light-controlled Ca²⁺ influx in neurons

In neurons, intracellular Ca²⁺ signals regulate a plethora of important physiological processes, action potentials and synaptic

plasticity over hundreds of seconds to neuronal development over days. Neuronal Ca²⁺ signals are therefore precisely controlled in space and time and recognized by a variety of Ca²⁺-binding proteins that direct the Ca²⁺ signal into specific biochemical pathways[47]. To evaluate if CapChRs allow for optogenetic initiation of Ca²⁺ signals in neurons on a fast time scale, we generated AAVs that express either mScarlet- or YFP-tagged CapChRs and C2-LC under the control of the neuron-specific human synapsin promoter (Fig. 5A). Co-expression with a cytosolic volume marker showed efficient membrane localization of all three opsins (Fig. S19). Since CapChR2-showed the most robust Ca²⁺-permeation in cell lines, we compared Ca²⁺ flux in cultured hippocampal neurons expressing CapChR2 or C2-LC. We ensured minimal cross-activation of the blue-light-sensitive opsins by choosing the 630 nm light-sensitive Ca²⁺ indicator Cal-630, which we applied via the patch pipette (Fig. 5B, C). This decision was based on prior results indicating that fluorescence acquisition with Fura-2 led to substantial cross-activation (Fig. S15B). Moreover, in order to minimize any secondary contribution of voltage- or ligand-activated Ca²⁺ channels, we recorded

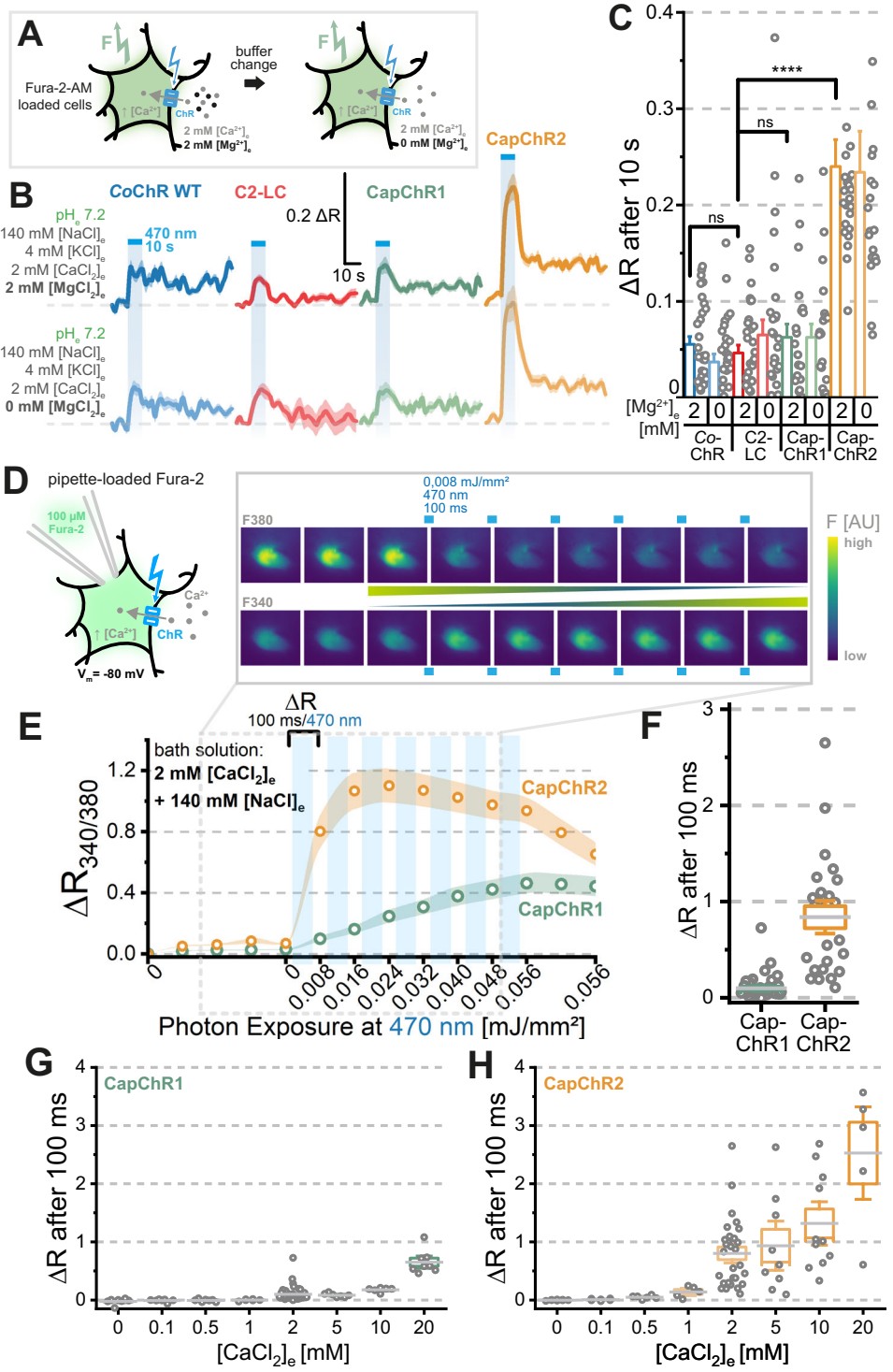

light-evoked Ca²⁺ signals at −70 mV in the presence of TTX and NBQX (and 1 mM [Mg²⁺]ₑ). Ca²⁺ signals were monitored over an 18-s period, in which C2-LC or CapChR2 were activated by a 4-s sequence of saturating blue light flashes (Fig. 5E, F). In CapChR2-expressing neurons, the 470 nm light flashes triggered a stronger Ca²⁺ indicator signal than in C2-LC-expressing neurons (C2-LC: peak $\Delta F/F = 0.78 \pm 0.15$, $n = 6$; CapChR2: peak $\Delta F/F = 2.19 \pm 0.41$, $n = 7$; Fig. 5E–G), whereas photocurrent amplitudes (Fig. 5I) and the total charge transfer were smaller in CapChR2-expressing neurons (Fig. 5H). Thus, CapChR2 provides less depolarization and more Ca²⁺ influx compared to C2-LC (Fig. 5I, J). For both opsins, blue light-evoked currents decayed with a similar time course (Fig. 5F).

## CapChRs robustly trigger calcium signalling in mammalian neurons

To evaluate the potential of CapChRs as genetically encoded calcium actuators (GECAs), we expressed them in mouse organotypic hippocampal slices. In this system, the cytoarchitecture is retained, and ex vivo genetic manipulations of the tissue can be performed[48]. Here, we used mScarlet-tagged CapChRs or C2-LC expressed by adeno-associated viruses (AAVs) (Fig. 6A). The three constructs were not cytotoxic and were uniformly expressed over all hippo-campal regions (example shown for CapChR2, Fig. 6A). We then evaluated the responses of principal neurons in whole-cell voltage clamp configuration upon illumination with blue light (see the

**Fig. 4 | Ca²⁺-flux at physiological levels of [CaCl₂]ₑ in ND7/23 cells. A–C** Fura-2-AM imaging on *Co*ChR WT (blue), C2-LC (red), CapChR1 (green) and CapChR2 (orange). **A** Overview of the experimental setup: Fura-2-AM loaded ND7/23 cells with a resting membrane potential were illuminated with 470 nm (-0.08 mW/mm², saturating light intensities) light to allow an influx of calcium ions through the expressed ChR in the presence and absence of bath Mg²⁺. **B** Mean imaging response of the denoted constructs under the two measured conditions (mean as coloured line and shadows represent SEM). **C** Quantified peak responses after 10 s of illumination. Single dots on the right of the columns represents one cell under those conditions (Bar: mean ± SEM). Number of replicates in (**B**) and (**C**) (+Mg²⁺/−Mg²⁺): *Co*ChR *n* = 30/30; C2-LC *n* = 32/30; CapChR1: *n* = 21/21; CapChR2: *n* = 24/21. Two-sided, unpaired Wilcoxon–Mann–Whitney-Test: \**P* ≤ 0.05, \*\**P* ≤ 0.01, \*\*\**P* ≤ 0.001, \*\*\*\**P* ≤ 0.0001. *P* values for comparison to C2-LC in **C**: *Co*ChR *P* = 0.35; CapChR1 *P* = 0.44; CapChR2 *P* = 5.07 × 10⁻¹⁰. **D–H** Voltage-clamped calcium imaging on CapChR1 and 2. **D** Experimental design for voltage-clamped measurements on ND7/23 cells. Cells were loaded with membrane-impermeable Fura-2 via the patch pipette (internal buffer: 110 mM [NaCl]ᵢ and divalent cation free, see the "Methods"

section). A baseline measurement was started for both the 380 and 340 nm channels (5 acquisitions), with subsequent 470 nm illumination (100 ms, -0.08 mW/mm², -0.008 mJ/mm² per $F_{340/380}$ ratio acquisition, saturating illumination for both CapChRs) to measure calcium influx through the CapChRs. A membrane voltage of −80 mV was applied at each illumination cycle. **E** Calcium imaging response (ratio of 340/380 nm fluorescence; $R_{340/380}$) for both CapChRs at physiological pH and ion concentrations. **F** Fluorescence change (Δ$R$) after 100 ms of illumination at −80 mV holding potential. **G** and **H** Fluorescence change (Δ$R$) for CapChR1 (green) and CapChR2 (orange) after 100 ms of illumination at −80 mV holding potential and at different extracellular calcium and sodium concentrations. Box middle line: Mean; Box outer edges ± SEM; Box whiskers: 1.5 × SEM. Number of replicates in **G** and **H**: CapChR1/CapChR2: 144 mM [NaCl]ₑ/0 mM [CaCl₂]ₑ: *n* = 7/8, 143.8 mM [NaCl]ₑ/0.1 mM [CaCl₂]ₑ: *n* = 7/6, 143 mM [NaCl]ₑ/0.5 mM [CaCl₂]ₑ: *n* = 7/6, 142 mM [NaCl]ₑ/1 mM [CaCl₂]ₑ: *n* = 6/5, 140 mM [NaCl]ₑ/2 mM [CaCl₂]ₑ: *n* = 45/29, 134 mM [NaCl]ₑ/5 mM [CaCl₂]ₑ: *n* = 7/9, 124 mM [NaCl]ₑ/10 mM [CaCl₂]ₑ: *n* = 6/11, 104 mM [NaCl]ₑ/20 mM [CaCl₂]ₑ: *n* = 8/5. *n* = X biologically independent cells. AU arbitrary units.

"Methods" section). With progressively increasing calcium concentrations ranging from 0 to 6 mM, inward currents from C2-LC became smaller when Mg²⁺ was replaced by Ca²⁺. On the contrary, in the case of both CapChR1 and CapChR2, the inward currents increased with [Ca²⁺]ₑ and an additional slow current with a large amplitude developed that extended beyond the light stimulus (Fig. 6B). In this context, CapChR2 especially stands out, since it elicits a light-triggered charge transfer almost 20-fold greater in presence of 6 mM calcium, compared to zero added calcium (Fig. 6C). Due to its high calcium permeability, the secondary currents observed in the case of CapChR2 suggest the activation of endogenous calcium-activated ion channels at the plasma membrane. However, neither small conductance SK nor high conductance BK calcium-activated K⁺ channels were responsible for the slow photocurrents, because we observed no inhibition during bath perfusion with either Apamin or Paxilline (Fig. 6D). In contrast, bath application of Niflumic Acid (NFA), which blocks calcium-activated chloride channels (CaCC)⁴⁹⁻⁵¹, caused a robust reduction of the inward current, abolished the slow tail current characteristic of CaCC in central neurons⁵², and typically reverted the photocurrent to a pure CapChR2 current (Fig. 6D). Under our recording conditions, the 10-fold chloride gradient across the plasma membrane should result in a Nernst potential of about −60 mV. We speculate that the CaCC, which allows this secondary inward current (chloride efflux), is potentially distant from the cell soma and thus possibly sees a reduced chloride gradient⁵³. Future work may reveal the biophysical or cell biological mechanisms underlying the slow tail current. For CapChR1, the effect of NFA was less obvious and non-existent for C2-LC (Fig. 6E), consistent with a negligible calcium entry for this mutant in our conditions.

The size and slow development of secondary photocurrents led us to ask whether calcium entry through CapChR2 could also trigger calcium-induced calcium release (CICR) from intracellular stores⁵⁴. Indeed, the application of Ryanodine, which inhibits this process in hippocampal neurons⁵⁵, caused a substantial reduction of the current, including the slow tail current. Co-applied NFA further reduced the current and abolished the long tail (Fig. 6D). Measurements in ND7/23 cells confirmed no effect of NFA on the CapChR2 current in isolation (Fig. S16). In current clamp measurements, depolarization due to CapChR2 activation led to robust spiking (Fig. 6F–H), but not in the presence of NFA, confirming that the photocurrent of CapChR2 alone is not enough to drive neuronal spiking. Thus, we conclude that CaCC is activated by calcium permeation through CapChR2, which is augmented by the store release of calcium (Fig. 6I). Taken together, these observations indicate CapChR2 is a superior calcium-conducting channelrhodopsin that robustly activates calcium-dependent processes in hippocampal neurons.

## CapChRs induce light-evoked calcium influx in *Drosophila* explant brain preparations

Neural circuit research in *Drosophila* is based on the cell-specific use of optogenetic reporters and actuators. To test the functionality of CapChR1 or CapChR2 in the *Drosophila* nervous system, we generated transgenic fly lines expressing CapChR1 or CapChR2 under the control of the UAS promoter. We targeted expression to the well-studied M4/6 mushroom body output neurons⁵⁶, while co-expressing the red-shifted fluorescent calcium indicator UAS-jRCaMP1a⁵⁷ (Fig. 7A). Calcium transients following blue light exposure were recorded from M4 dendrites of intact explant brains at an extracellular calcium concentration of 1.5 mM (Fig. 7B). We next probed different illumination times, while comparing the efficacy of CapChR2 to that of CapChR1 and *Cr*ChR2. Whereas C2 failed to elicit any visible calcium entry similarly to control cells, CapChR1 induced a small calcium increase after 2 s blue illumination (Fig. 7C, D). However, in CapChR2-expressing neurons, pulses of only 50 ms duration prompted a substantial calcium increase, while 500 ms pulses caused a saturating calcium response that was not further increased by prolonged illumination. Notably, CapChR2 induced a slight fluorescence ramp-up from the beginning of the recording, likely caused by CapChR2 activation with the green imaging light (Fig. 7C). These data confirm the specific action of CapChR2 to raise cytosolic calcium in targeted cell types, in an intact brain.

## Discussion

We have characterized the calcium conductance and selectivity of several CCRs and mutant derivatives. The insights gained into channelrhodopsin variants have allowed us to better understand their calcium permeation and engineer derivatives of *Cr*ChR2 and *Co*ChR with enhanced calcium selectivity, which we designated CapChR1 and CapChR2, respectively. CapChR2 represents a potent calcium actuator for the optical dissection of calcium-dependent processes in vertebrate and invertebrate host systems.

To enable further discussion and to identify the molecular constraints that are responsible for the higher Ca²⁺ preference in *Co*ChR and the voltage-dependent Ca²⁺ conductance for both CapChRs, we performed molecular dynamics (MD) simulations, based on the available C2 crystal structure⁴⁰, coupled with p$K_a$ calculations to account for protonation changes. The resulting equilibrated structures suggest that the carboxyl group of E258 in CapChR1 is oriented towards the extracellular cavities, whereas D63 points inward, with water permeating up to the central gate and close to the RSB in the closed state (Fig. S17). In this orientation, E258 could recruit and strengthen Ca²⁺-binding at the access channel, assisting the inward permeation of divalent cations. In CapChR2, both D43 and E238 carboxyl groups point towards the outer pore at the end of the equilibration, suggesting superior recruitment and stabilization of calcium ions with the

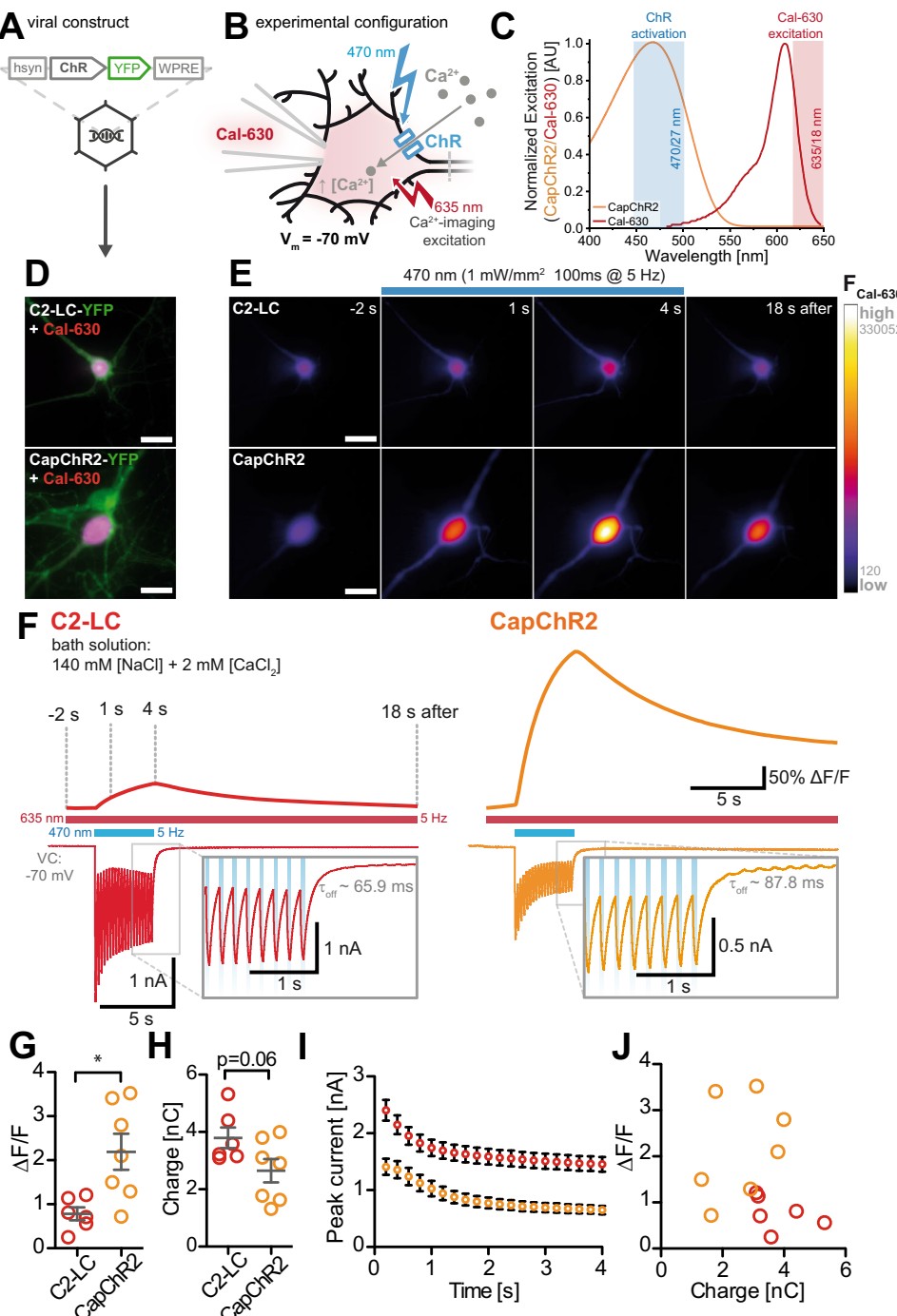

**Fig. 5 | Quantification of light-driven Ca²⁺ flux in neurons. A** Design of the AAV-expression vector with YFP fused to the opsins. **B** Whole-cell voltage-clamp experiments allowed direct perfusion of the Cal-630 indicator, with high loading efficiency in the soma. **C** Exemplary activation spectra of CapChR2, the excitation spectrum of Cal-630 and the band-pass filtered light for ChR and Cal-630 excitation. CapChR2 action spectra reproduced from Supplementary Fig. 10, with $n = 6$, the excitation spectrum of Cal-630 provided by the manufacturer. **D** Exemplary live-cell wide-field images of neurons expressing C2-LC-YFP or CapChR2-YFP, overlaid with a maximum intensity projection of the Cal-630 signal obtained during the recording session (scale bars: 10 µm). **E** Fluorescent images of the Cal-630 signal before (−2 s), after 1 or 4 s of blue light flashes, or 18 s after blue-light illumination from the cells shown in **C** (scale bars: 10 µm). **F** Single-trial intensity

profiles of somatic Ca²⁺ signals evoked by 20 50-ms flashes of 470 nm (1 mW/mm², saturating conditions) at 5 Hz in a C2-LC (red) or CapChR2 (yellow)-expressing neuron. Simultaneously recorded currents are shown underneath. Insets show the current decay after the last light flash. **G** Comparison of the maximum $\Delta F/F0$ evoked by blue-light flashes (C2-LC, $n = 6$; CapChR2, $n = 7$; unpaired, two-tailed $t$-test: $p = 0.012$). **H** Quantification of the total charge transfer evoked by the opsin stimulation (C2-LC, $n = 6$; CapChR2, $n = 7$ unpaired, two-tailed $t$-test: $p = 0.063$, no adjustment for multiple comparisons). **I** Time course of peak currents evoked by 470 nm flashes in C2-LC and CapChR2-expressing neurons (C2-LC, $n = 6$; CapChR2, $n = 7$). **J** Scatter plot of the light-evoked charge transfer against the maximum Ca²⁺ signal within single cells. Graphs depict the mean ± SEM. *$P \le 0.05$, **$P \le 0.01$, ***$P \le 0.001$, ****$P \le 0.0001$. $n = X$ biologically independent cells. AU arbitrary units.

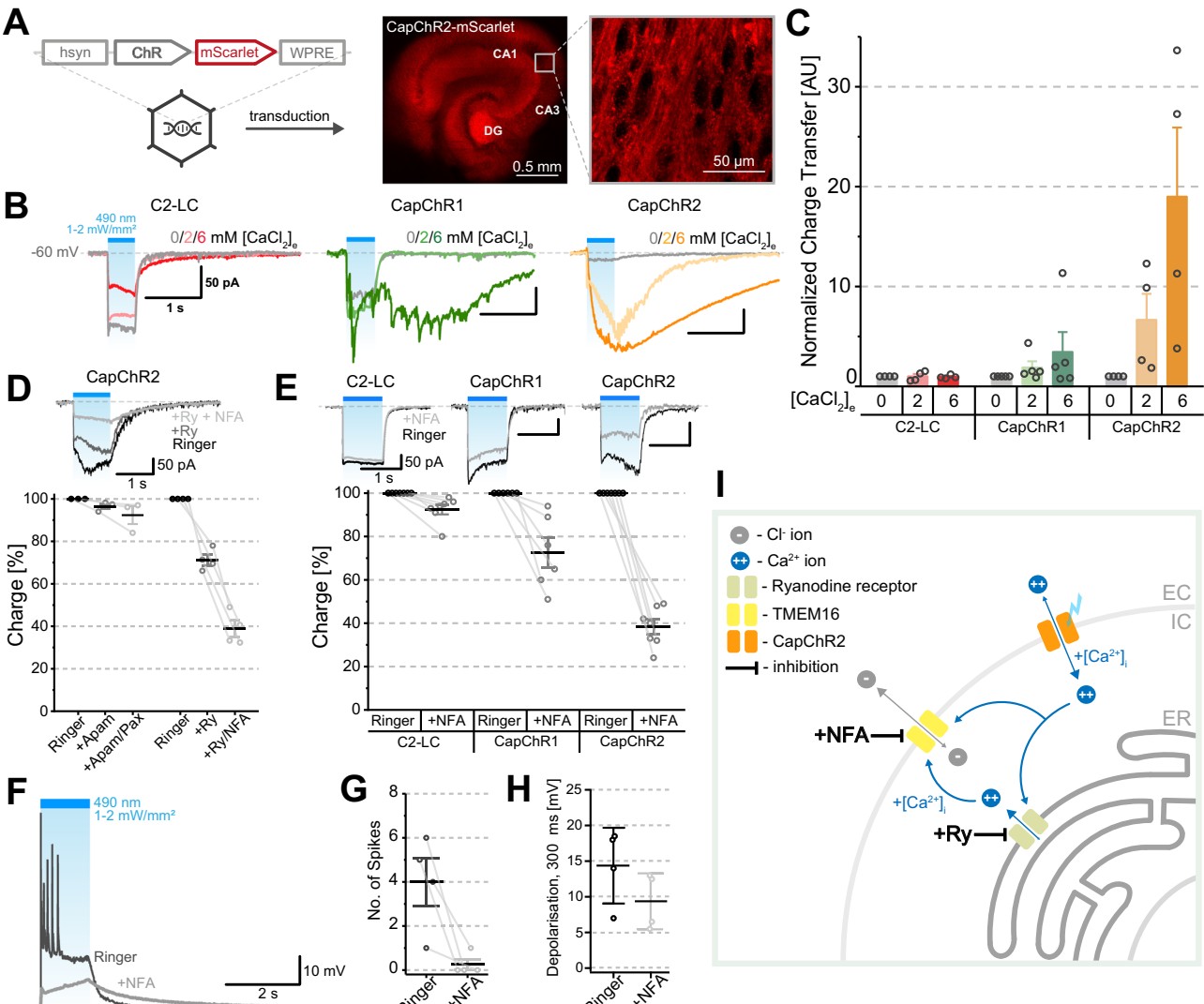

**Fig. 6 | Electrophysiological characterization of CapChRs in organotypic hippocampal slice culture. A** Construct design and fluorescent confocal micrographs showing organotypic hippocampal cultures (DIV 20) transduced with AAVs-expressing mScarlet-tagged ChRs. All three constructs C2-LC, CapChR1 and CapChR2 were homogeneously expressed throughout the hippocampal tissue after 3–4 days post-transduction (example shown for CapChR2). **B** Representative whole cell current traces depicting the effects of 500 ms of 490 nm (1–2 mW/mm², saturating) light exposure at different calcium concentrations. **C** Charge transfer (normalized to values at 0 mM [CaCl2]e) between 1 and 2 s of illumination ($n = 4$ for C2-LC; $n = 5$ for CapChR1; $n = 4$ for CapChR2). **D** Photocurrents from CapChR2 in presence of blockers and the total charge transfer in % of the control. Apamin (Apam; 10 µM) Paxilline (PAX; 10 µM) ($n = 3$), Ryanodine (Ry, 50 µM) ($n = 4$) and Niflumic Acid (NFA, 0.5 mM, $n = 4$). **E** Effects of NFA application on photocurrents of C2-LC ($n = 7$), CapChR1 ($n = 6$) and CapChR2 ($n = 7$). **F** Exemplary whole cell voltage traces of CapChR2-expressing neurons, depicting the effects of 1 s of 490 nm light exposure in current clamp mode in the presence or absence of NFA in the bath solution. **G** Number of spikes elicited during illumination in (**F**) (mean, ±SEM; $n = 4$ cells). **H** Summary of the effect of NFA on pyramidal cell depolarization (mean, ±SEM; $n = 4$). **I** Schematic representation of the physiological response in CapChR2-expressing slices. Calcium entry via CapChR2 elicits calcium store release and activates calcium-activated chloride channels (TMEM16). Graphs depict the mean ± SEM. $n = X$ biologically independent cells. AU arbitrary units, EC extracellular, IC intracellular, ER endoplasmic reticulum.

help of these two amino acids (Figs. 2C, S17). The orientation of the carboxylic residues provides an explanation for the strong Ca²⁺ inward, but not outward conductance, exhibited by both CapChRs. It also provides a possible reason for the calcium-dependent off-kinetics (Fig. 3I): the coordination of D43 and E238 close to the RSB enables strong Ca²⁺ binding, which might slow down structural changes required for channel closing at high calcium concentrations.

Due to a lack of an open-state crystal structure, the precise permeation of cations through ChRs has not been elucidated to date. Notwithstanding, we performed molecular dynamics (MD) simulations on homology models that are based on the crystal structure of dark-adapted C2[40] to obtain insights into possible Ca²⁺-uptake mechanisms in CapChR2 (see "Methods" section for details). Dark-state MD

simulations were performed in a high Ca²⁺ environment with an external electric field applied across the membrane (Fig. S20). Under these conditions, Ca²⁺ was bound to the extracellular protein surface for several picoseconds via the carboxylic residues E81, D83 and D84 that deprotonate more easily in the presence of the divalent cation due to a lowered p$K_a$ (Fig. S20A). Out of these interactions, only Ca²⁺ binding to E81 initialized Ca²⁺-uptake (Figs. 8A, S20B). Upon an E81 sidechain rotation, the ion was transferred to the deprotonated E77 (Fig. 8B), which already showed a low p$K_a$ due to its proximity to positively charged K73 and R100 (Fig. S20B 1–3). In silico, for 4–5 ns, the ion was held in between both glutamates (Figs. 8B, S20C left). Nevertheless, although a second Ca²⁺ had already bound to E81 (Figs. 8B, S20B-4, C right), the driving force was not sufficient to

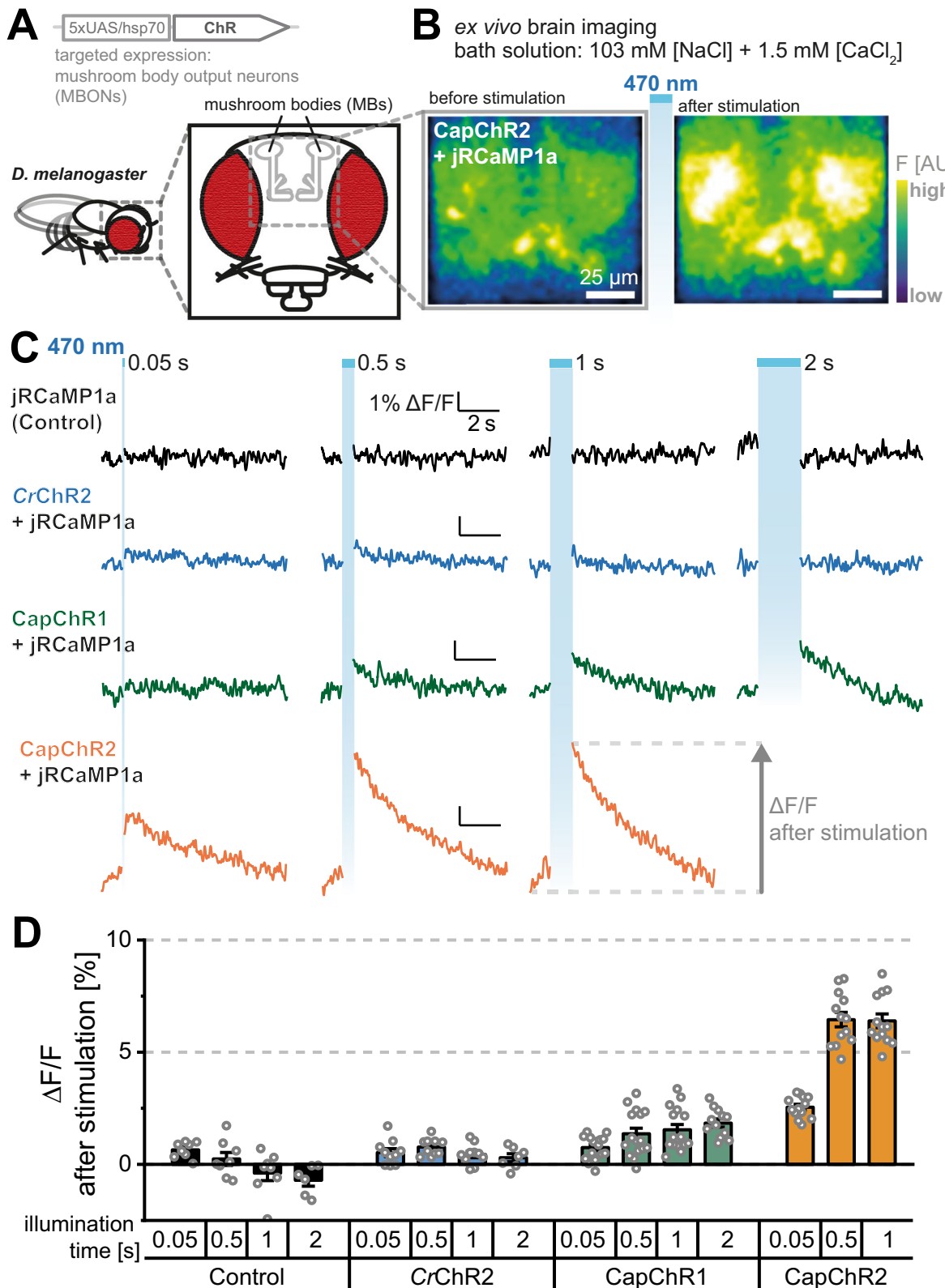

**Fig. 7 | Targeted activation of CapChR2 reliably elicits calcium influx to single *Drosophila* neurons. A** Expression of UAS-CapChR1, UAS-CapChR2 and UAS-*Cr*ChR2 was targeted to M4/6 mushroom body output neurons using the VT1211-Gal4 driver line. **B** Blue-light (470 nm, ~4.5 mW/cm²) activation of CapChR2 elicits marked calcium increase in M4/6 neurons, measured with co-expressed jRCaMP1a in *Drosophila* explant brain preparations. **C** Example jRCaMP1a fluorescence traces following blue-light pulses of indicated duration (blue bars). **D** Fluorescence signal quantification following blue-light stimulation. For 0.05 s, 0.5 and 1 s of blue light excitation *n* = 16 for CapChR1, *n* = 13 for CapChR2, *n* = 10 for *Cr*ChR2 (C2) and *n* = 8 for the control. For 2 s blue light excitation *n* = 12 for CapChR1, *n* = 7 for *Cr*ChR2 and *n* = 6 for the control. Bar charts depict the mean ± SEM. *n* = X biologically independent flies. AU arbitrary units.

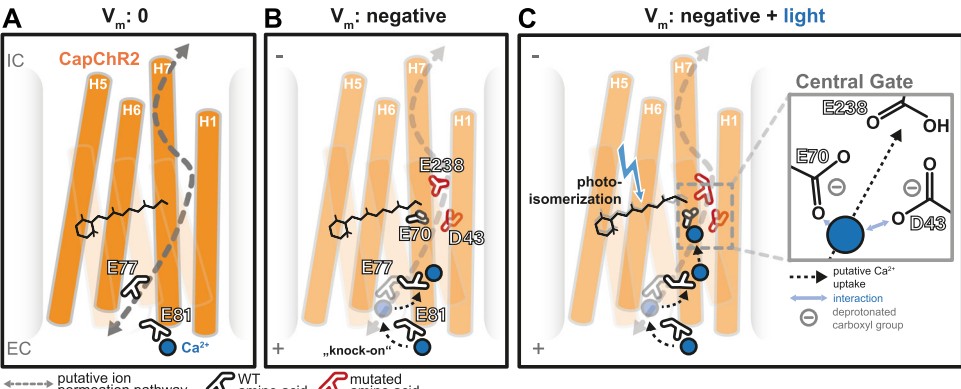

**Fig. 8 | Proposed model for Ca²⁺-permeation and inward rectification in CapChR2 based on MD simulations. A** Stable binding of Ca²⁺ to E81 in the absence of membrane voltage ($V_m$) and no illumination. **B** Upon the application of a negative membrane potential, a second Ca²⁺ ion binds to E81 and the first ion is displaced into the pore ("knock-on"), where it binds E77 and flips towards the central gate.

**C** Isomerization of the retinal allows the deprotonation of E70 and D43 with the help of Ca²⁺, which interacts with both carboxyl groups. Further along the putative permeation pathway is E238, which might assist in continued passage through the pore. These interactions are only possible when ions permeate from the extracellular side (EC) to the intracellular side (IC).

support further ion permeation closer to the RSB area in this first simulation with an applied electrical field, and Ca²⁺ rested stably at E77.

The next possible interaction partners in the pore at this point were D43, E70 and E238, all pointing toward the extracellular side. p$K_a$ calculations on residues at the central gate showed that E70 was deprotonated, while both D43 and E238 were consistently protonated. It is conceivable that this constellation serves as a barrier for Ca²⁺ translocation in darkness. For continued ion translocation, D43 or E238 deprotonation might be needed to overcome the distance to the intracellular part of the pore. To enable further ion uptake and increase the electrostatic pull for the Ca²⁺ ion, we forced deprotonation of D43 after 35 ns of Ca²⁺ binding to E77 in a second simulation. After a few ns, both bound Ca²⁺ ions were consecutively transported towards the central gate, in what can be described as a "knock-on"-mechanism (Figs. 8, S20B 4–5 and D). Despite the general lack of accuracy for single-atom Ca²⁺ models in conventional MD, this observation supports the idea that, upon retinal isomerization, the p$K_a$ of D43 or E238 is possibly lowered, leading to deprotonation and increased cation attraction (Fig. 8C). Compared to S43 and N238 in the parental *Co*ChR, this sudden drop in electric charge would decrease the energy barrier for Ca²⁺-transfer past the RSB and, hence, increase its conductance drastically. This mechanism is in line with the experimentally observed and here further outlined inward rectification of the Ca²⁺ photocurrent at a negative voltage since the orientation of participating residues is skewed towards the extracellular side. In addition, there were uncharged polar amino acids along the proposed translocation pathway that interacted shortly with the bound Ca²⁺, such as Q29, N33, Q36 and Y223. These residues could contribute to cation stabilization in general. In conclusion, our MD simulations provide theoretical insight into our experimental observations of an inward rectifying calcium selectivity in CapChR2. The observed interactions suggest the orientation of relevant residues towards the extracellular access channel, enabling enhanced and unidirectional selectivity for Ca²⁺ during ion translocation from the extracellular to the intracellular side.

Our measurements support previous claims that ChR variants with increased calcium conductance (when compared to C2) do not strongly discriminate between H⁺, Na⁺, Ca²⁺ and Mg²⁺[17]. Moreover, ChRs are known to possess voltage-dependent ion binding[16], cation selectivity[17] and kinetics[58] as well as voltage-dependent ion competition. To date, the latter was mainly discussed for H⁺ and Na⁺, based on the crossing of $I(E)$-curves for different extracellular H⁺/Na⁺ ratios[41,59]. In fact, previous studies suggested that the use of the Goldman–Hodgkin–Katz (GHK) equation for conductance ratios is

insufficient to model ion permeation through wild-type *Cr*ChR2[16]. Amongst others, models with a single, asymmetric barrier were created, which described the inward rectifying $I(E)$ data better than the GHK model. As shown in our measurements (Figs. 1, S2, S3, S7), C2 poorly conduct Ca²⁺ in comparison to Na⁺, but the aforementioned voltage-dependent barrier for the Ca²⁺ conductance can be massively reduced by modification of the C2 central gate residues, primarily S63D and N238E in CapChR1, which are similar to selectivity filters in other calcium channels (Figs. S5, S17)[42]. However, the Ca²⁺ conductance remains voltage-dependent with little conductance for the divalent cation close to 0 mV. Along similar lines, the improved Ca²⁺ selectivity of CapChR2 seems to be related to the already reduced H⁺ conductance in the wild-type protein (Fig. S2) in conjunction with the putative calcium-binding site at the central gate proposed in our model (Fig. 8). This inward-rectified and voltage-dependent Ca²⁺-selectivity does not correlate with the properties of other calcium channels (e.g. CRAC channels[60]). Taking into account the data obtained from our MD simulations, it is possible that the asymmetric barriers in CapChR2 are the carboxylic residues oriented towards the extracellular side (Fig. 8).

When expressed in mammalian cells, naturally occurring CCRs predominantly conduct H⁺ and Na⁺ under physiological conditions. As a result, using these constructs as calcium actuators in this environment is futile, since depolarization initiated by the strong influx of H⁺ and Na⁺ causes cell depolarization, opposing further Ca²⁺ influx due to the reduced driving force and the strong voltage-dependence of Ca²⁺ permeation. Moreover, in excitable animal tissue, depolarization induces action potentials and secondary activation of endogenous voltage-gated calcium channels, which in turn trigger myriad downstream processes caused by both membrane depolarization and cytosolic calcium. Therefore, discrimination of cause-and-effect of action potential firing and elevations in [Ca²⁺]ᵢ has been impracticable. In mouse neurons, we observed that CapChR2 triggered a much-enhanced Ca²⁺ signal at physiological concentrations compared to C2-LC without substantial differences in decay kinetics (Fig. 5). This observation suggests that Ca²⁺ conducted by CapChR2 is the major source of the recorded Ca²⁺ increase and underscores its usefulness as a GECA with high temporal resolution. Importantly, as proof-of-principle for functional use, CapChRs trigger calcium-dependent physiological responses upon illumination, including calcium-induced calcium release from stores and secondary activation of calcium-activated chloride channels, which we discriminated pharmacologically (Fig. 6). We therefore conclude that CapChR2 presents the possibility for more versatile experiments, separating calcium

influx from neuronal excitation. We have used wide-field illumination in these experiments, but patterned illumination, perhaps coupled to subcellular targeting, should further improve spatial resolution for the dissection of calcium signalling in eukaryotic cells. This could, for example, allow studies on synaptic plasticity in subcellular compartments such as specific dendritic branches or clusters of spines with high spatiotemporal precision.

We also demonstrate that CapChR2 is a valuable addition to the *Drosophila* optogenetic toolbox. Similar to mammalian preparations, we observe CapChR2-induced transients displaying slow decay kinetics in agreement with a largely calcium-dominated activation of the studied neurons. Although we cannot exclude the activation of endogenous voltage-gated calcium channels in our *Drosophila* experiments, this activity should be observable in C2 and CapChR1 measurements (Fig. 7). Thus, the difference between CapChR1 and CapChR2 should be an indicator for the superior performance of the latter independent of endogenous channels. However, the possibility of calcium-induced calcium release from intracellular stores remains, which would evidence the efficient initiation of calcium signalling. In *Drosophila*, it should be specifically interesting to use this tool to directly manipulate memory programs that have been previously shown to depend on specific intracellular calcium signalling in behaving flies[61]. Likewise, changing intracellular calcium levels have been tied to the animal's sleep status in neurons of the central complex and glial cells of flies[62,63]. CapChR2 now provides the technical means to investigate the underlying physiological mechanics and directly manipulate calcium pathways shaping an animal's behaviour. The strong light-dependent activation of calcium-induced calcium release in CapChR2-expressing mammalian neurons signifies the potency of the channel for the activation of endogenous calcium signalling. This lends credence to our assumption that CapChR2 is a powerful optogenetic tool to manipulate intracellular calcium concentrations with high spatiotemporal precision in tissues containing competing ions in the extracellular environment.

CapChRs represents a major step forward in understanding and improving calcium binding and permeation in ChRs. Being entirely genetically encoded, CapChRs should permit non-invasive optical manipulation of calcium levels in tissue and intact whole organisms with a high temporal resolution. We hope that they empower experimenters with the tools necessary to dissect calcium-signalling processes with minimal invasion, high precision and without crosstalk.

## Methods

Experiments involving mice were carried out according to the guidelines stated in directive 2010/63/EU of the European Parliament on the protection of animals used for scientific purposes and were approved by the local authorities in Berlin, Germany. Animals were therefore maintained in compliance with the EU Legislation on the protection of animals used for scientific purposes and were approved by Landesamt für Gesundheit und Sociales Berlin (LaGeSo).

### Molecular biology

For expression in mammalian cells, human codon-optimized sequences for *Cr*ChR2, C1V1, C1C2 and *Ps*ChR were cloned into pmCherry-N1 as previously described[3,64]. DNA encoding for ReaChR was also subcloned as previously described[65]. Briefly, they were sub-cloned in-frame via XbaI/BamHI (*Cr*ChR2, C1V1), HindIII/BamHI (C1C2), NheI/AgeI (*Ps*ChR) or HindIII/XbaI (ReaChR) restriction sites into either pmCherry-N1 (*Cr*ChR2, C1V1, C1C2 and *Ps*ChR) or pmCerulean3-N1 (ReaChR). bReaChEs-TS-eYFP-ER and ChRmine-TS-eYFP_ER were provided by Karl Deisseroth and cloned by us via Gibson assembly[66] into pCDNA3.1. Cheriff has been ordered from Addgene (#51697) and subsequently cloned into pmScarlet-C1. *Co*ChR and *Ts*ChR were synthesized and subsequently cloned into pmScarlet-C1. ChroME was generated by site-directed mutagenesis on Chronos and cloned into

pmScarlet-C1. Site-directed mutagenesis of *Cr*ChR2, *Ps*ChR, *Ts*ChR and *Co*ChR was performed via Quikchange using a *Pfu* polymerase (Agilent Technologies, Santa Clara, CA) with the primers denoted in Supplementary Data 1.

For neuronal expression, coding sequences of C2-LC, CapChR 1 and 2 were fused to either mScarlet or YFP and inserted into an AAV expression vector (backbone of Addgene Plasmid #58806) between the neuron-specific human synapsin promoter and the Woodchuck-Hepatitis-Virus Posttranscriptional Response Element (WPRE). AAV particles (serotype 9) were provided by the Viral Core Facility (VCF) of the Charité Berlin.

### Phylogenetic analysis

For phylogenetic analysis, selected channelrhodopsin sequences were aligned using mafft (G-INS-i mode) v. 7.479[67]. Using the alignment, the phylogenetic tree was generated with iqtree v. 1.6.12[68,69] with automatic model selection and 1000 ultrafast bootstrap alignments. The tree was adjusted using interactive Tree of Life (iTOL) v. 6.3.3[70]. A separate sequence alignment was generated for selected channelrhodopsins using ClustalW[71]. Sequence Alignments were aligned and plotted using ClustalX[71] in the Jalview[72] software.

### Mammalian cell culture

ND7/23 cells (Sigma-Aldrich, St. Louis, MO, USA) were grown on glass coverslips coated in poly-D-lysine, which were placed in 35 mm Petri dishes containing Dulbecco's modified Eagle medium (DMEM; Biochrom GmbH) with 5% (v/v) foetal bovine serum (FBS superior; Biochrom, Berlin Germany), glutamine (Biochrom, Berlin, Germany) and 100 μg/ml penicillin/streptomycin (Biochrom, Berlin, Germany). Moreover, growth media contained 1 μM all-*trans* Retinal. To transiently transfect cells, 6 μl FuGENE HD transfection reagent (Promega, Madison, WI) was incubated with 2 μg of vector DNA in 250 μl DMEM for 15 min and added to the cells 2 days prior to measurements.

### Calcium imaging with Fura-2-AM in ND7/23 cells

The growth medium was supplemented with 2 μM Fura-2-acetoxymethyl ester (Life Technologies GmbH, Darmstadt, Germany) 45 min, prior to measurements. The excitation light, sourced from an Optoscan Monochromator (Cairn Research, Kent, UK), was coupled to the optical path of an inverted microscope Olympus IX70 (Olympus, Tokyo, Japan) equipped with an Olympus ×40 water immersion objective (Olympus, Tokyo, Japan) and a FF493/574 dichroic mirror (AF-Analysetechnik, Tübingen, Germany) for fluorescence imaging with a pco.panda 4.2 sCMOS camera (Kelheim, Germany). For ChR activation, the Optoscan Monochromator was operated at a centre wavelength of $470 \pm 5$ nm for 10 s with an intensity of ~0.08 mW/mm$^2$. For Fura-2 excitation, consecutive 50 ms exposure flashes of $340 \pm 10$ nm ($0.02$ mW/mm$^2$) and $380 \pm 10$ nm ($0.14$ mW/mm$^2$) were applied with the Optoscan Monochromator. The sampling rate of the Fura-2 signal was 0.5 Hz. The bath solution used for Fura-2-AM imaging contained 1 mM [NaCl], 1 mM [KCl], 1 mM [CsCl], 2 mM [MgCl$_2$], 70 mM [CaCl$_2$] and 10 mM [HEPES], adjusted to pH 7.2 and 320 mOsm using glucose.

### Whole-cell patch-clamp recordings in ND7/23 cells

Transfected ND7/23 cells were measured at 24 °C with fire-polished, borosilicate patch pipettes (Science Products GmbH, Hofheim, Germany) pulled with a P-1000 micropipette puller (Sutter Instrument, Novato, USA). Pulled pipettes had an access resistance of 1.5−3 MΩ. Membrane resistance was generally >0.5 GΩ, and access resistance was generally <10 MΩ. A pE-4000 CoolLED (CoolLED, Andover, UK) was used for excitation. The CoolLED light path was coupled into an Axiovert 100 TV inverted microscope (Carl Zeiss, Oberkochen, Germany), which was used to search for fluorescent cells through a ×40/1.0 water objective (Carl Zeiss, Oberkochen, Germany).

**Table 1 | Buffer compositions used in experiments on cultured ND7/23 cells**

| | Concentration in mM | | | | | | | | Osmolarity [mOsm] | pH | Figure reference |
|---|---|---|---|---|---|---|---|---|---|---|---|
| | NaCl | KCl | CsCl | CaCl$_2$ | MgCl$_2$ | HEPES/Tris | EDTA | NMGCl | | | |
| High [CaCl$_2$]$_e$ | 1 | 1 | 1 | 70 | 2 | 10 | 0 | 0 | 320 | 7.2 | Figs. 1, 2, S2–S4, S6–S8 |
| High [NaCl]$_i$ | 110 | 1 | 1 | 2 | 2 | 10 | 10 | 0 | 290 | 7.2 | Figs. 2, S2–S4, S6–S8, S14 |
| High [NaCl]$_e$ | 140 | 1 | 1 | 2 | 2 | 10 | 0 | 0 | 320 | 7.2 | Figs. 2, S2–S4, S6–S9, S13 |
| High [NMGCl]$_e$ | 1 | 1 | 1 | 2 | 2 | 10 | 0 | 140 | 320 | 7.2 | Figs. 2, S2–S4, S6–S8 |
| High [CaCl$_2$]$_e$ | 1 | 1 | 1 | 70 | 2 | 10 (Tris) | 0 | 0 | 320 | 9 | Figs. 2, S2–S4, S6–S8 |
| High [MgCl$_2$]$_e$ | 1 | 1 | 1 | 2 | 70 | 10 | 0 | 0 | 320 | 7.2 | Figs. 2, S9, 14 |
| DVCF: high [NaCl]$_i$ | 110 | 1 | 1 | 0 | 0 | 10 | 0.05 | 0 | 290 | 7.2 | Figs. 3, 4, S10–S13, S15, S16 |
| 144 mM [NaCl]$_e$, 0 mM [CaCl$_2$]$_e$ | 144 | 1 | 1 | 0 | 0 | 10 | 0 | 0 | 320 | 7.2 | Figs. 3, 4, S10–S13, S15, S16 |
| 140 mM [NaCl]$_e$, 2 mM [CaCl$_2$]$_e$ | 140 | 1 | 1 | 2 | 0 | 10 | 0 | 0 | 320 | 7.2 | Figs. 3, 4, S12, 15 |
| 134 mM [NaCl]$_e$, 5 mM [CaCl$_2$]$_e$ | 134 | 1 | 1 | 5 | 0 | 10 | 0 | 0 | 320 | 7.2 | |
| 124 mM [NaCl]$_e$, 10 mM [CaCl$_2$]$_e$ | 124 | 1 | 1 | 10 | 0 | 10 | 0 | 0 | 320 | 7.2 | |
| 0 mM [NaCl]$_e$, 70 mM [CaCl$_2$]$_e$ | 1 | 1 | 1 | 70 | 0 | 10 | 0 | 0 | 320 | 7.2 | Figs. 3, S10, 12, 14 |
| 142 mM [NaCl]$_e$, 1 mM [CaCl$_2$]$_e$ | 142 | 1 | 1 | 1 | 0 | 10 | 0 | 0 | 320 | 7.2 | Figs. 4, S15 |
| 143 mM [NaCl]$_e$, 0.5 mM [CaCl$_2$]$_e$ | 143 | 1 | 1 | 0.5 | 0 | 10 | 0 | 0 | 320 | 7.2 | |
| 143.8 mM [NaCl]$_e$, 0.1 mM [CaCl$_2$]$_e$ | 144 | 1 | 1 | 0.1 | 0 | 10 | 0 | 0 | 320 | 7.2 | |
| 104 mM [NaCl]$_e$, 20 mM [CaCl$_2$]$_e$ | 104 | 1 | 1 | 20 | 0 | 10 | 0 | 0 | 320 | 7.2 | |
| DVCF: 144 mM [NMGCl]$_e$, 0 mM [CaCl2]$_e$ | 1 | 1 | 1 | 0 | 0 | 10 | 0 | 144 | 320 | 7.2 | S12, 13 |
| 140 mM [NMGCl]$_e$, 2 mM [CaCl$_2$]$_e$ | 1 | 1 | 1 | 2 | 0 | 10 | 0 | 140 | 320 | 7.2 | S12 |
| 134 mM [NMGCl]$_e$, 5 mM [CaCl$_2$]$_e$ | 1 | 1 | 1 | 5 | 0 | 10 | 0 | 134 | 320 | 7.2 | |
| 124 mM [NMGCl]$_e$, 10 mM [CaCl$_2$]$_e$ | 1 | 1 | 1 | 10 | 0 | 10 | 0 | 124 | 320 | 7.2 | |
| | NaCl | KCl | CsCl | CaCl$_2$ | MgCl$_2$ | HEPES | TEA | NMGCl | | | |
| High [KCl]$_e$ | 1 | 140 | 5 | 2 | 2 | 10 | 20 | 0 | 320 | 7.2 | Figs. 2, S9 |
| | NaCl | KCl | CsCl | BaCl$_2$ | SrCl$_2$ | HEPES | MgCl$_2$ | NMGCl | | | |
| 144 mM [NaCl]$_e$, 2 mM [MgCl$_2$]$_e$ | 144 | 1 | 1 | 0 | 0 | 10 | 2 | 0 | 320 | 7.2 | S13, S14 |
| 144 mM [NaCl]$_e$, 2 mM [BaCl$_2$]$_e$ | 144 | 1 | 1 | 2 | 0 | 10 | 0 | 0 | 320 | 7.2 | S13 |
| 144 mM [NaCl]$_e$, 2 mM [SrCl$_2$]$_e$ | 144 | 1 | 1 | 0 | 2 | 10 | 0 | 0 | 320 | 7.2 | S13 |
| | NaCl | KCl | CsCl | CaCl$_2$ | MgCl$_2$ | HEPES/Tris | EDTA | NMGCl | | | |
| 140 mM [NaCl]$_e$, 2 mM [MgCl$_2$]$_e$ | 140 | 4 | 0 | 2 | 2 | 10 | 0 | 0 | 320 | 7.2 | 4A–C |
| 144 mM [NaCl]$_e$, 0 mM [MgCl$_2$]$_e$ | 144 | 4 | 0 | 2 | 0 | 10 | 0 | 0 | 320 | 7.2 | 4A–C |

The signal was amplified using an ELC-03XS amplifier (npi Electronic, Tamm, Germany) digitized using a Digidata 1440A (Molecular Devices, Sunnyvale, CA). An agar-enveloped AgCl electrode containing 140 mM NaCl was submerged in the bath solution and served as a reference electrode. A perfusion system (Ringer-Bath-Handler, Lorenz Meßgerätebau, Germany) was used for buffer exchange. ChR activation was performed using the 470 nm channel of the CoolLED system with an intensity of ~1.9 mW/mm². Ion selectivity measurements were performed using a protocol with 500 ms LED flashes to induce currents, which were recorded from −80 to +40 mV in 20 mV steps. Cells were pre-illuminated for 1 s before measurements with 1–2 min waiting time before initializing protocols. For action spectra, low-intensity light, sourced from a Polychrome V (TILL Photonics, Planegg, Germany), ranging from 390 to 680 nm was flashed in 10-nm steps for 10 ms at −60 mV. Photon irradiance was kept equal through all wavelengths via a motorized neutral-density filter wheel (Newport, Irvine, CA), which was controlled by custom LabVIEW software (National Instruments, Austin, TX). Light titrations were performed using the CooLED system eliciting currents for 1 s and at −60 mV at the denoted light intensities. Recordings were automated and performed using the Clampex software suite (Molecular Devices), which was also used to pre-correct liquid junction potentials. The intracellular solution used in Figs. 2, S2–4, S6–8, S14 contained: 110 mM NaCl, 1 mM KCl, 1 mM CsCl, 2 mM CaCl$_2$, 2 mM MgCl$_2$, 10 mM HEPES and 10 mM EDTA adjusted to pH 7.2 and 290 mOsm. The intracellular solution used in Figs. 3, 4, S10–13, S15, S16 contained: 110 mM NaCl, 1 mM KCl, 1 mM CsCl, 10 mM HEPES and 0.05 mM EDTA adjusted to pH 7.2 and 290 mOsm. All buffers used in experiments are listed in Table 1.

## Voltage-clamped Fura-2 calcium imaging in whole-cell configuration

ND7/23 cells were patched and recorded as described in "Whole-cell patch-clamp recordings in ND7/23 cells". However, membrane resistance was generally >1 GΩ and access resistance was <5 MΩ. This was done to remove any false-positive $\Delta R$ caused by a calcium leak. After a patch was established and the cell was ruptured, 5 min waiting time was included to ensure optimal loading of the calcium indicator. 100 µM Fura-2-pentasodium salt (Life Technologies GmbH, Darmstadt, Germany) was included in the internal buffer for fluorescence recordings of calcium influx shortly before measurements. An Optoscan Monochromator (Cairn Research, Kent, UK) served as the light source, with a beamline coupled to an Olympus IX70 (Olympus, Tokyo, Japan) equipped with an Olympus ×40 water immersion objective (Olympus, Tokyo, Japan) and an FF493/574 dichroic mirror (AF-Analysetechnik, Tübingen, Germany) for fluorescence imaging with a pco.panda 4.2 sCMOS camera (Kelheim, Germany). For ChR activation,

the Optoscan Monochromator was operated at a centre wavelength of $470 \pm 5$ nm for 100 ms with an intensity of -0.08 mW/mm². For Fura-2 excitation, consecutive 250 ms exposure flashes of $340 \pm 8$ nm (-0.02 mW/mm²) and $380 \pm 8$ nm (-0.12 mW/mm²) were applied with the Optoscan Monochromator with a holding potential of 0 mV. During activation of the ChR, the membrane voltage was held at −80 mV. Measurements were synchronized using Clampex (Molecular Devices) and μ-Manager[73]. The intracellular solution was a $Ca^{2+}$-free buffer containing: 110 mM NaCl, 1 mM KCl, 1 mM CsCl, 10 mM HEPES and 0.05 mM EDTA adjusted to pH 7.2 and 290 mOsm (with added 100 μM Fura-2-pentasodium salt). All buffers used in experiments are listed in Table 1.

## Expression in human embryonic kidney cells
Genes encoding *Co*ChR and CapChR2 were cloned into a pcDNA3.1 vector in a frame with a C-terminal Kir2.1 membrane-trafficking signal (KSRITSEGEYIPLDQIDINV) and a 1D4 affinity tag for protein purification. Fusion constructs were expressed under the control of a CMV-promotor in Human Embryonic Kidney cells (HEK293T-cells) that were cultured in Dulbecco's modified medium (DMEM) with stable glutamine supplemented with 10% (v/v) foetal bovine serum, 1 μM all-*trans* retinal, and 100 μg ml⁻¹ penicillin/streptomycin (SIGMA), seeded in 10 cm Corning® Cell Culture dishes and transfected using the PEI method[74].

## Protein purification from HEK cells
HEK293T was harvested 2 days post-transfection at a confluence of >90% and subsequently washed with Dulbecco's phosphate-buffered saline (DPBS; Gibco, Thermo Fisher Scientific), containing 2X complete protease inhibitor (Merck, Darmstadt, Germany) as described before[75]. Cell pellets were resuspended in purification buffer containing 150 mM NaCl, 3 mM MgCl₂ and 50 mM HEPES at pH 6.5 and solubilized with 1.5% DDM/0.3% CHS in the presence of 5 μM all-trans-retinal for 4 h at 4 °C. Insoluble material and cell debris were separated by centrifugation at $45,000 \times g$ for 30 min at 4 °C. 150 μl 1D4 affinity beads (CUBE Biotech GmbH, Monheim, Germany) were added to the supernatant and incubated at 4 °C overnight. Beads were washed three times in an ultrafree-MC spin column with purification buffer supplemented with 0.03% DDM/CHS. Bound proteins were eluted in two steps with washing buffer, supplemented first with 0.4- and then with 1 mg/ml 1D4 peptide (GeneScript, NL). The purified protein was finally transferred to either 70 mM CaCl₂, 20 mM HEPES, pH 7.0 or 100 mM NaCl, 20 mM HEPES, pH 7.0 both supplemented with 0.03% DDM/CHS and absorption spectra were recorded using a Shimadzu UV-2000 photospectrometer with UVProbe v2.34. Protein samples were illuminated for 30 s with a single 455 nm LED at light intensities of 2.55 mW cm⁻².

## Primary hippocampal neuron culture
Mice used for the experiments lived in standardized conditions of temperature (18–23 °C) and humidity (40–60%). Hours of light and darkness (12-h light/12-h dark cycle) were maintained at a constant level all year round. Hippocampi were dissected from P0 to P1 C57/BL6-N mice of either sex, and cells were dissociated by papain digestion followed by manual trituration. Neurons were seeded on glial feeder cells at a density of $1.57 \times 10^4$ cells/cm² in 24-well plates and maintained in Neurobasal-A supplemented with 2% B27 and 0.2% penicillin/streptomycin (Invitrogen).

## Electrophysiology and imaging in primary neurons
After 1–3 days in vitro, AAV particles encoding ChR-YFP were added to the cultured neurons to a final concentration of $1.74 \times 10^{10}$ particles/ml. Light-evoked currents and $Ca^{2+}$ signals were recorded at room temperature on an inverted microscope (IX73 Olympus) after 15–21 days in vitro. Fluorescent light (470 and 630 nm) from an LED (pE-4000, CoolLED) was passed through an appropriate filter (DAPI/FITC/Cy3/

Cy5 Quad sbx LED HC Filter Set, AHF) and a ×60 water objective (UPLSAPO60XW/1,2 U Plan S Apo). Fluorescence signals were captured with a sCMOS camera (Photometrics Prime BSI Express) using the Micromanager software[73]. Cells were voltage clamped at −70 mV using a Multiclamp 700B amplifier under the control of a Digidata 1550 AD board and Clampex 10 software (Molecular Devices), which was also used to trigger the LED and camera. The extracellular solution contained (in mM): 140 NaCl, 2.4 KCl, 10 HEPES, 10 glucose, 2 CaCl₂ and 1 MgCl₂ (pH 7.3, 300 mOsm). Imaging and optical stimulation were performed in the presence of 1 μM TTX, 5 μM NBQX and 4 μM Gabazine. The intracellular solution contained (in mM): 127.3 K-Gluconate, 19.6 KCl, 9.8 HEPES, 3.9 Mg-ATP, 12.5 Phosphocreatine-Na, 0.3 Na-GTP. On the day of the recording, 50 μg of Cal-630 potassium salt (AAT Bioquest) was freshly dissolved in 300 μl of the intracellular solution, to a final concentration of 146 μM. The sequence of Cal-630 imaging and 470-nm light stimulation was initiated 3 min after achieving the whole-cell configuration. Recordings were discarded if holding currents exceeded −150 pA. Camera exposure time was 25 ms, with $2 \times 2$ binning. Cal-630 was excited with 630 nm at 10 mW/mm², and channelrhodopsins were stimulated with 50 ms flashes of 470 nm light at 1 mW/mm² (all intensities recorded above objective).

## Confocal imaging
At DIV 1, primary neurons grown in 24-well plates were transduced with $6 \times 10^9$ AAV particles (serotype 9) encoding either ChR2(LC)-mScarlet, CapChR1-mScarlet, CapChR2-mScarlet, or soluble mScarlet alone. All constructs were expressed under the control of the neuron-specific human synapsin promoter (hsyn). In addition, $1 \times 10^{10}$ particles of an AAV encoding hSyn:EGFP (serotype 9) was added to each well. At DIV 21, samples were fixed for 10 min with 4% paraformaldehyde at room temperature, washed 2× with PBS and 1× with H₂O before mounting. Single plane confocal images were acquired using a Leica TCS SP5 with the 561 and 451 nm laser lines and an HCX PL APO ×63 NA 1.4 oil objective with a zoom of 2, image size set to $2048 \times 2048$.

## Preparation, electrophysiology and immunocytochemistry on organotypic hippocampal slices
Mice used for the experiments lived in standardized conditions of temperature (18–23 °C) and humidity (40–60%). Hours of light and darkness (14-h light/10-h dark cycle) were maintained at a constant level all year round. Organotypic hippocampal cultures were prepared as previously described[48]. Briefly, 350 μm-thick transverse sections from P6 to P9 WT C57BL/6J mouse (Jackson Laboratory) hippocampus were prepared on filter paper following the interface method procedure and grown in a MEM-based mouse slice culture medium in an incubator with 5% CO₂ at 34 °C. 3–5 μl of AAVs containing different constructs were added 4–6 days before the day of recording by adding the AAV solution to the culture medium. Somatic whole cell patch clamp recordings of visually identified transduced CA principal cells were performed at room temperature (22–24 °C) after 16–23 days of slice culture with microelectrodes (3–8 MΩ tip resistance) prepared from borosilicate glass capillaries (1B150F-4 World Precision Instruments) using a Sutter P-1000 puller. Slices were superfused with recirculating aCSF (2–3 mL/min) containing 145 mM NaCl, 2.5 mM KCl, 2 mM CaCl₂, 1 mM MgCl₂, 10 mM HEPES, 10 mM glucose, adjusted to 310/315 mOsm/l and pH of 7.3 with NaOH. The intracellular solution contained 135 mM CH₃KO₃S, 4 mM NaCl, 2 mM MgCl₂, 10 mM HEPES, 2 mM Na₂ATP, 0.3 mM Na₂GTP, 0.06 mM EGTA and 0.01 mM CaCl₂, adjusted to 300 mOsm/l and pH 7.3.

Voltage clamp recordings were performed in presence of bath-applied Tetrodotoxin (TTX, 1 μM; Tocris); by holding neurons at −60 mV (uncorrected junction potential value was calculated to be −6.6 mV). To monitor the series resistance (<20 MΩ), a hyperpolarizing voltage step (−10 mV, 100 ms) was delivered at the beginning of each recording and recordings with series resistance changes >20% were

 

discarded. Current clamp recordings were done under bridge balance compensation, holding neurons at −60 mV in presence of extracellular GYKI 52466 (100 μM; AbCam), AP-5 (20 μM; HelloBio), and SR-99531 (10 μM; HelloBio). Simultaneous detection of mScarlet-positive neurons (red emitting) and optogenetic activation of different constructs was achieved by using a 405/488/594 nm Laser Triple Band filter set (TRF 69902; Chroma/AHF) mounted in a Zeiss TIRF cube (Dichroic filter: ZET 405/488/594m-TRF, Emission filter: ZT 405/488/594rpc-UF2; Chroma/AHF). Red and Blue light was generated by LEDs (490 nm centre wavelength, Thorlabs M490L4) and directed through a Zeiss epifluorescence reflector (Examiner A1) into the water immersion ×60 objective (Olympus LUMPlanFLN; N.A. 1). Data was acquired using a Multiclamp 700B amplifier (Molecular Devices), digitized at 20 kHz under the control of the Axograph software (Axograph Scientific) and analysed with Axograph and IgorPRO 8 (WaveMetrics). Immunocytochemistry on hippocampal cultures was performed by fixing the slices right after the electrophysiological experiments for one hour at RT in 0.4% paraformaldehyde (PFA), in phosphate buffer solution (PBS), then mounted on glass coverslips using Immo-Mount (Fisher Scientific) and imaged with a laser scanning confocal microscope (Zeiss LSM780; ×10 objective).

### *Drosophila* rearing and cloning
Flies were raised on standard cornmeal food and kept at 65% humidity, 25 °C and a 12 h light–dark cycle. The driver line used was VT1211-Gal4[56]. The CapChR2 sequence was cloned into a pUAStB vector. Transgenic flies were created by Bestgene (CA, USA) and insertions were targeted into the VK0005 landing site on the third chromosome. Other expressed genetic constructs were UAS-CapChR2, UAS-*Cr*ChR2 (Bloomington-No. 9681) and UAS-jRCaMP1a[57].

### *Drosophila* fluorescence microscopy
Brain preparations were essentially performed as previously described[76]. Brains of 5–12-day-old flies of mixed sexes were dissected on ice. The flies expressed UAS-jRCaMP1a or UAS-jRCaMP1a together with either UAS-CapChR1, UAS-CapChR2, or UAS-*Cr*ChR2 under the control of VT1211-Gal4. The head capsule was removed and placed into carbogenated solution containing 103 mM NaCl, 3 mM KCl, 5 mM N-Tris, 10 mM trehalose, 10 mM glucose, 7 mM sucrose, 26 mM NaHCO$_3$, 1 mM NaH$_2$PO$_4$, 1.5 mM CaCl$_2$, 4 mM MgCl$_2$, 295 mOsm, pH 7.3. Fat tissue and trachea were removed from the brain using fine forceps. Brains were imaged with an Olympus MX51WI wide-field microscope with a ×40 Olympus LUMPLFLN objective. jRCaMP1a was excited using green light (550 nm, 0.5 mW). Brain recordings were performed 3 times and lasted for 10 s. During each recording, a blue light (470 nm, 3.5 mW or ~4.5 mW/cm$^2$) pulse was applied after 2 s. The blue light pulse had a length of 0.05 s in the first, 0.5 s in the second and 1 s in the third recording. For CapChR1, *Cr*ChR2 and the control an additional recording with a 2 s light pulse applied after 2 s was taken after 1–5 min. Videos were obtained at 20 Hz temporal resolution with an Andor iXON Ultra camera controlled by Solis software. Recordings and blue light pulses were controlled using the Axon Digidata 1550B and the PClaMP 10 Software. A Lumencor Spectra X-Light LED engine served as the source of blue and green light. Videos were analysed using NOSA v1.1, Excel365 and Origin.

### System preparation for MD simulations
The 3D models of *Co*ChR, CapChR1 and CapChR2 were generated based on the *Cr*ChR2 crystal structure (PDB-ID 6EID, https://doi.org/10.2210/pdb6EID/pdb[40]). For CapChR1, the *Cr*ChR2 structure was directly modified while for CapChR2, a homology model of *Co*ChR (amino acids 12–258) was computed using SWISS-MODEL[77] and used as a template. Each protomer of *Co*ChR was equipped with a retinal cofactor bound to K237 using PyMOL version 2.5.0 (PyMOL−The

PyMOL Molecular Graphics System, Version 2.0 Schrödinger, LLC.). Respective mutations (S63D, L132C, T159C and N258E for CapChR1, S43D, L112C, T139C and N238E for CapChR2) were introduced using CHARMM version c42b1[78]. Internal hydration was adopted from the *Cr*ChR2 crystal structure. For equilibration, all systems containing the protein and internal waters were introduced into a homogeneous DMPC bilayer membrane, respectively, surrounded by water and neutralized using 0.15 M KCl using CharmmGUI[78]. For ion conductance simulations, the CapChR2 system was neutralized using 0.15 M CaCl$_2$ instead.

### Classical MD simulations in CHARMM
MD simulations on *Cr*ChR2, *Co*ChR, CapChR1 and CapChR2 were carried out using CHARMM version c42b1 and openMM version 7.0rc1[79]. The MD simulation was computed under NPT conditions using a 2 fs time step, a 303.15 K Langevin heat bath, the particle-mesh Ewald method for long-range electrostatics, and the CHARMM36 force field[80]. Before and in between the initial equilibration steps, the protonation of all titratable residues was adjusted based on Karlsberg2+ predictions[81,82]. For further production runs, combined p$K_a$-MD simulations were computed: during classical MD simulations, p$K_a$ values were calculated periodically for 11 snapshots of each 10 ns production run using Karlsberg2+MD[82] to subsequently adapt the protonation pattern of the system pH dependently (pH 8).

### Ion conductance simulations in NAMD
Voltage-dependent MD simulations on CapChR2 were carried out using NAMD version 2.14[83]. The equilibration and the first 1 ns of the production steps were computed similarly to the CHARMM calculations without voltage. Subsequent steps were continued with a voltage of −1 V applied over the z-axis of the system using the NVT ensemble to avoid large pressure fluctuations. p$K_a$ calculations and protonation state adjustments were applied similarly to the CHARMM calculations, but in 5 ns intervals and for a pH of 7.5. After 80 ns of simulation, the protonation state of CapChR2-D43 was manually defined to enforce its deprotonation.

### p$K_a$ calculations in Karlsberg2+
In Karlsberg2+, the holoprotein structure and ions within a 4 Å cutoff were explicitly included. The rest was substituted with continuum solvation and an implicit ion concentration of 100 mM. Single-structure p$K_a$ calculations during the equilibration were performed using APBS[84] and three pH-adapted conformations (PACs), while Karlsberg2+MD used the extracted snapshots and created only single PACs at pH 7. All PACs were generated using Karlsberg2+ in a self-consistent cycle including adjustment of protonation patterns of titratable amino acids and salt bridge opening according to either pH −10, 7, or 20.

### Data analysis
Data were analysed using Clampfit 10.4 (Molecular Devices) and Origin 2017 (OriginLab, Northampton, MA). Stationary photocurrents were used for the determination of ion selectivity and light sensitivity, which were calculated by averaging the photocurrents in the last 50 ms of illumination. Action spectra and light titrations were normalized to the maximal photocurrents. A three-parameter Weibull function was employed to estimate the peak of action spectra. Reversal potentials were extrapolated by linear regression of the 2 data points closest to 0 pA. τ-values for off-kinetic calculations were obtained via a bi-exponential fit of the currents at −80 mV if not indicated otherwise. Structural data were analysed and depicted using CueMol 2.2, with the single exception of Fig. S20, which was produced using PyMol 2.5.0. Statistical analysis for ND7/23 datasets was performed using the Estimation Stats online tool[85].

## Statistics

Statistical significance between the two groups was determined by using either a two-sided unpaired Wilcoxon–Mann–Whitney test or a two-sided two-group *t*-test. For statistical significance within a single group, a paired two-sided Wilcoxon–Mann–Whitney test was used. Statistical tests used are denoted in the corresponding figure legends and outputs from the Estimation Stats online software[85] are included in the source data.

## Reproducibility

Immunostaining in the right panel of Fig. 6a was repeated three times with similar results. Confocal imaging in Supplementary Fig. 19 was repeated two times with similar results.

## Data availability

CapChR plasmids CapChR1_pmCherry-N1 (https://www.addgene.org/188031/), CapChR2_pmScarlet-N1 (https://www.addgene.org/188032/), pAAV-Syn_CapChR1_eYFP (https://www.addgene.org/188033/), pAAV-Syn_CapChR2_eYFP (https://www.addgene.org/188035/), pAAV-Syn_CapChR1_mScarlet (https://www.addgene.org/188037/), pAAV-Syn_CapChR2_mScarlet (https://www.addgene.org/188039/), CapChR1_pUAST-attb (https://www.addgene.org/188040/) and CapChR2_pUAST-attb (https://www.addgene.org/188041/) used in this study can be found on Addgene. The data generated in this study used for graphs are provided in the Source Data file. Source data are provided with this paper. The structural data cited in this study are available online under PDB ID: 6EID, https://doi.org/10.2210/pdb6EID/pdb and PDB ID: 4MVQ, https://doi.org/10.2210/pdb4mvq/pdb. The sequences used in Supplementary Fig. 1 can be found under the following accession codes numbers: CrChR2 (AF508966), CoChR1 (AHH02107), C1C2 (PDB: 3UG9, https://doi.org/10.2210/pdb3UG9/pdb), ReaChR (KF448069.1), C1V1 (KF448069.1), PsChR (JX983143.1), TsChR (AHH02155), Chronos (KF992040.1), ChRmine (QDS02893), and Chrimson (AHH02126). Source data are provided with this paper.

## Code availability

The code used for MD simulations will be provided upon request.

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

## Acknowledgements

We thank Sandra Augustin, Maila Reh and Tharsana Tharmalingam for their excellent technical assistance. We also thank Karl Deisseroth, Charu Ramakrishnan, Brian Hsueh and Peter Wang for initial testing in Cardiomyocytes and a ChRmine-TS-eYFP_ER plasmid. Additionally, we would like to thank Kai R. Konrad and Rainer Hedrich for initial testing in plants. The work was supported by the German Research Foundation (DFG): TRR186 (Grant No. 278001972, P.H. and A.J.R.P.), SFB1078 (Grant No. 221545957, P.H.), SPP1926 (Grant No. 425994138, P.H. and B.R.R.), EXC-2049 (Grant No. 390688087, D.O., P.H., A.P., D.S., J.V.), SFB1315 (Grant No. 327654276, D.O., P.H., D.S., J.V., B.R.R.), Heisenberg Professorship (Grant Nos. 323514590, 446182550, A.J.R.P.), Emmy Noether Programm (D.O.), by the Hector Fellow Academy (HFA, Grant No. 30000619, P.H.), and by the European Research Foundation (Grant No. 767092 "Stardust", P.H., ERC synergy grant GA 810580 to D.S.). N.P.P. was the recipient of an EMBO Long-Term Fellowship (no. ALTF 873-2018). P.H. is a Hertie Professor for Neuroscience and is supported by the Hertie Foundation. D.S. is an Einstein Professor and is supported by the Einstein Foundation.

## Author contributions

R.G.F.L., P.H. and A.J.R.P. developed the project with contributions from all authors. R.G.F.L., N.P.P., M.-M.H., L.T., J.V and B.R.R. did experimental work, with assistance from J.W. R.G.F.L. and L.T. performed electrophysiological characterization in ND7/23 cells. B.R.R. performed measurements on primary neurons. N.P.P. performed measurements on organotypic slices. M.-M.H. performed measurements on *Drosophila*. J.V. performed spectroscopic characterization. E.S. performed computational analysis. Data analysis was performed by R.G.F.L., N.P.P., M.-M.H., L.T., J.V., B.R.R., E.S. and J.O. and interpretation was done by all authors. R.G.F.L., P.H., A.J.R.P., D.O., D.S. and B.R.R. wrote the paper, with contributions from all authors.

## Funding

## Competing interests

The authors declare no competing interests.
