## [Peer Review File · Nature Communications]

Calcium permeable channelrhodopsins for the photocontrol of calcium signallingREVIEWER COMMENTS

Reviewer #1 (Remarks to the Author):

Summary:

Aiming for ChR variants with higher calcium selectivity the authors combined mutations, which have been implicated in increasing CrChR2 Ca²⁺-conductance. The comparison of different CrChR2 combinatory mutants by patch-clamp experiments yielded the calcium-permeable CrChR2 quadrupole mutant CapChR1. The transfer of the corresponding set of mutations to CoChR, which the authors selected based on a calcium imaging assay, led to the calcium-permeable ChR variant CapChR2. CapChRs exhibit a high Ca²⁺ conductance at negative voltages. The authors point out utility of CapChRs as genetically encoded calcium actuators in ND7/23 cells, in primary hippocampal neuron cultures, in organotypic hippocampal slices and in M4/6 mushroom body output neurons from *Drosophila* brain explants.

Major comments:

p1 line 22, p.8 lines 177-185, Fig. S4

“...the resulting CoChR (Co-SD-LC-TC-NE) expressed well with reliable membrane localization. It exhibited improved calcium permeation at negative holding potentials, similar to C2-SD-LC-TC-NE (Fig. 2D and H). Recordings on Co-SD-LC-TC-NE also revealed a high relative calcium permeation ($I_{Ca} / I_{Na} \sim 1.9$), a positive Erev-shift in calcium buffer in comparison to the WT, and low proton permeability (Fig. 2J and K, S2H). According to these properties, we designated C2-SD-LC-TC-NE and Co-SD-LC-TC-NE as CapChR1 and CapChR2 respectively (Calcium permeable ChannelRhodopsins)...”

Fig. S4 shows that the Erev-shift upon replacement of external sodium by calcium and the relative calcium permeation at -80 mV (I_{Ca}/I_{Na}) are likely not significantly different in CapChR1, CapChR2 and ChR2 S63D (Plazzo et al., 2012). The authors included the previously described S63D mutation into their quadruple mutants CapChR1 and CapChR2. Hence, the authors description of CapChRs having “strongly improved Ca²⁺ permeation” (p. 1 line 22) is not sufficiently supported by their results. A potentially beneficial property of CapChR2 is its higher photocurrent density (Fig S4 C), which may result from good plasma membrane expression. Please perform significance tests for the comparison of the photocurrent densities. The authors should, as outlined in the subsequent comments, provide further evidence for CapChRs superiority to prior existing genetically-encoded calcium actuators.

Fig. 3 D,E,F, Fig S4 C, p.4 lines 89 to p.5 line 103, p.10 lines 219-226

Measuring the difference of the photocurrent reversal potentials at varying ionic conditions is of key importance for relative cation permeability determination (e.g. P_{Ca}/P_{Na}). The authors state that “For C2-LC, we confirmed large inward photocurrents in presence of 140 mM [NaCl]_e and decrease in amplitude upon replacement of Na⁺ by Ca²⁺ (Figure 3A, D and G), indicating poor conductance for Ca²⁺. This conclusion is also supported by the negative Erev-shift (ΔE_{rev}) upon complete sodium replacement (Fig. 3D). Conversely, both CapChRs exhibit a calcium-dependent increase of inward currents (Fig. 3B and C). At -80 mV, currents increased ~3-fold for CapChR1 (Fig. 3E and H) and ~5-fold for CapChR2 when Na⁺ was replaced with Ca²⁺, coupled with minimal impact on the reversal

potentials for CapChR2 (Fig. 3F and I).” These statements are questionable, because the variability of the measured reversal potentials, in particular in the presence of 70 mM calcium, is much higher for CapChR2 (Fig. 3 F) than the variability of the measured reversal potentials for CapChR1 (Fig. 3 E) and C2-LC (Fig. 3 D). A likely reason for the observed variability is voltage drifts in the performed patch-clamp experiments. Therefore additional experiments for CapChR1 and CapChR2 reversal potential determination at 144 mM [NaCl]_e, 5 mM [CaCl₂]_e + 134 mM [NaCl]_e and 70 mM [CaCl₂]_e should be performed (using solution exchange) which include measurements at identical ionic conditions for voltage drift control. The authors could then exclude measurements, which show considerable voltage drifts. The high variability of the measured reversal potentials is also apparent in Fig. S4 C, which compares the calcium permeation of previously described ChR variants. The authors may corroborate their findings by additional measurements as outlined above. Alternatively, the corresponding paragraph (p.4 lines 89 to p.5 line 103) should be carefully rewritten.

Figure 1 D, Figure 4, p. 4 lines 77-79 and p. 12 lines 263-266, Figure 5, p. 20 lines 473-478

By employing the Fura-2 and Cal-630 calcium imaging assays the authors quantify intracellular Ca²⁺ accumulation while the intracellular solution is continuously exchanged against the solution in the patch pipette. Thereby intracellular calcium accumulation not only depends on the calcium permeabilities of the ChR variants but also on e.g. the rate of diffusional exchange between cells and patch pipette, on the cellular surface to volume ratio and on ChR plasma membrane expression. The inset in Figure 1 D should therefore not be labeled with “Ca²⁺-permeability”. Due to the outlined limitations of the calcium imaging assay, it is questionable if sentences like “However, both CoChR and TsChR showed larger calcium conductance than C2-WT and similar to C2-LC (Figure 1C and 78 D).” and “In mouse neurons, we observed that CapChR2 triggered a 3-fold stronger Ca²⁺ signal at physiological concentrations compared to C2-LC without substantial differences in decay kinetics (Figure 5).” are accurately describing the results. The authors should also address the question if the blue light pulses, which were applied in the calcium imaging assays evoked saturating or subsaturating photocurrents in the investigated ChR variants (e.g. Fig. 5, photocurrents of C2-LC-YFP and CapChR2-YFP evoked by 50 ms flashes of 470 nm light at an intensity of 1 mW/mm²).

Minor Comments:

p.2 lines 33-35

“Several natural and engineered ChRs have been shown to be selective for H⁺, Na⁺ or Cl⁻–7...”

The engineered ChRs, which are described in the given references show considerable permeation of other cations in addition. Therefore, changing the sentence to e.g. “...engineered ChRs have been shown to exhibit a higher selectivity for H⁺, Na⁺ or Cl⁻–7...” would provide a more precise description.

p.4 lines 88-89

“However, in most of these cases, Na⁺ vs. Ca²⁺ permeation has not been rigorously examined.”

Please clarify by referencing for which of the reexamined ChR variants Na⁺ vs. Ca²⁺ permeation has been examined before and for which of the reexamined ChR variants Na⁺ vs. Ca²⁺ permeation has not been examined before.

p. 5 lines 118-123, p. 7 lines 151-153, p. 19 line 452 – p. 20 line 456, p. 18 line 401 – p. 19 line 433, Fig. 8

“Varying E90 mutations modified either the Na⁺/H⁺ ratio or even converted C2 into an anion channel^{4,38}. Similarly, homotetrameric Na⁺-selective channels have been converted into Ca²⁺-channels by introduction of negative charges to their selectivity filters^{39,40}...”

“Based on our initial voltage-clamp experiments (Fig. S3), and following the logic outlined above, additional negative charge in the proximity of E90 might further increase relative Ca²⁺ permeation.”

“... the aforementioned voltage-dependent barrier for the Ca²⁺ conductance can be massively reduced by modification of the CrChR2 central gate residues, primarily S63D and N238E in CapChR1, which are similar to selectivity filters in other calcium channels (Fig. S14) 39.”

The authors repeatedly mention similarities between the modified central gate in CrChR2 mutants and selectivity filters of calcium channels. That interesting hypothesis may eventually be supported by a supplementary figure showing a high-resolution structure based comparison of the calcium channel selectivity filter and the central gate of CrChR2. The MD simulation provides limited insights in that regard, because it is based on the closed channel structure and does not directly address the potential role of the CapChR mutations in calcium permeation.

Figure S7, Fig. S15, p. 8 line 186-194

“CapChR2 shows a 4 nm red-shift and band narrowing at high [CaCl₂]_e, a possible indicator for a Ca²⁺-binding site close to the protonated retinal Schiff base (RSBH⁺) (Fig. S7). To analyse the impact of the mutations on the spectral properties of CoChR and CapChR2, we recorded absorption spectra from purified proteins (Fig. S15). We observed no absorption shift in CapChR2 compared to the parental WT and no change upon Ca²⁺ supplementation. We presume that the small spectral changes seen in the action spectra upon replacement of Na⁺ with Ca²⁺ might only be present at negative membrane voltages.”

Please prove the significance of the observed 4 nm red-shift (Fig. S7), in particular as this finding is not in line with the recorded absorption spectra. The presumption that spectral changes are voltage dependent should be corroborated by measurements of action spectra at different membrane potentials. Alternatively, the corresponding paragraph may be deleted.

Figure 6, p. 15 lines 347-350, p. 16 lines 359-361

“...under our recording conditions, the 10-fold chloride gradient across the plasma membrane should result in a Nernst potential of about -60 mV. The CACL-channels that allow this secondary inward current (chloride efflux) are potentially distant from the cell soma and thus possibly see a reduced chloride gradient⁴⁹.”

“In current clamp measurements, depolarization due to CapChR2 activation led to robust spiking (Fig. 6F-H), but not in the presence of NFA, confirming that the photocurrent of CapChR2 alone is not enough to drive neuronal spiking.”

The finding that even though the Nernst potential for chloride was about -60 mV, CACL-channel activation by CapChR2 led to depolarizing currents, which were essential to drive neuronal spiking, is explained by the localization of CACL channels distant from the soma, where local chloride gradients shift the Nernst potential to more positive voltages. Please provide a reference that shows that CACL-channels are expressed predominantly distant to the soma in hippocampal neurons.

Figure 7, p. 17 lines 377-392

Based on experiments in *Drosophila* M4/6 mushroom body output neurons using jRCaMP1a, a GECI that is used for neural activity imaging, the authors conclude that “These data confirm the specific action of CapChR2 to raise calcium in targeted cells types, in an intact brain.”. Please discuss the contribution of voltage activated calcium channels on the one hand and the contribution of CapChR2 on the other hand to the observed increase of the intracellular calcium concentration in the *Drosophila* M4/6 mushroom body output neurons.

Significance:

ChR variants, which show a strongly increased Ca²⁺-selectivity are an important extension of the optogenetic toolbox. Remote control of intracellular Ca²⁺ concentration by a light-activated genetically encoded Ca²⁺ actuator is of great interest for cell biology, because a large number of cell biological processes depend on Ca²⁺. CapChRs will potentially prove useful for the investigation of Ca²⁺ signaling processes.

Reviewer #2 (Remarks to the Author):

The manuscript “Channelrhodopsin-based calcium actuators for the photocontrol of calcium signalling” by Fernandez Lahore et al. is a very important work on the rational design of novel Channelrhodopsins (ChRs) that allow selective conducting of Ca²⁺ ions by illumination. In fact, researchers from various disciplines applying optogenetic tools including myself are awaiting such ChR2 desperately because Ca²⁺ is the most important intracellular second messenger involved in all cells of the body. Until to date, optogenetic tools for direct influx of Ca²⁺ from the extracellular space physiologically containing ~ 2 mM Ca²⁺ are lacking and the best available tool CatCh has been characterized at extracellular non-physiologic levels of 70 mM Ca²⁺.

In the present manuscript, the authors initially recapitulate previous Ca²⁺ imaging data with very high (70 mM) external Ca²⁺ and additionally show that wild-type ChR from other species (CoChR, TsChR) are similarly Ca²⁺ conducting than CatCh, which is derived from CrChR2 (Fig. 1). Subsequently, the authors nicely explain the rational design of a better Ca²⁺ permeable ChR by combining ChRs from different species with previously known mutations promoting Ca²⁺ entry and ChRs kinetics (LC, TC) with novel mutations (NE, SD) at the inner central gate (Fig. 2 and many supplemental Figures towards this goal). This resulted in ChRs with quadruple mutations termed “Calcium permeable ChannelRhodopsins” (CapChR1 and 2, but clearness of abbreviations is discussed below).

CapChR1 and 2 were analyzed in depth by electrophysiology (Fig. 3) and, in contrast to CatCh, showed larger currents in 70 mM Ca²⁺ than in 140 mM Na⁺, indicative of Ca²⁺ over Na⁺ selectivity. Because of the surprisingly strong inward rectification allowed Ca²⁺ currents only at very negative membrane potentials (-60 and -80 mV) Ca²⁺ imaging was performed while patch clamping cells at -

80 mV. In these key experiments CapChR2 showed large Ca²⁺ influx even at physiological Ca²⁺ concentrations of 2 mM (Fig. 4) but unfortunately Ca²⁺ imaging experiments in the presence of Mg²⁺ or direct comparison to CatCh was not performed. Finally, the authors nicely applied CapChRs in neurons (Fig. 5), organotypic hippocampal slices (Fig. 6) and *Drosophila* brain preparations (Fig. 7) to show that CapChR2 outperforms CatCh. Finally, using MD simulations the authors try to explain Ca²⁺ conductance and the rectification in CapChRs with a novel mechanism of ChRs gating (Fig. 8).

The results are original, important, timely and of high significance to the optogenetic field and related fields applying optogenetics. The conclusions are based on the extensive data, the methodology is sound with sufficient details explained (with an exception on intracellular solution, see below) and the work meet the expected standards in the field.

I highly recommend publication with minor revision and some more Ca²⁺ imaging experiments showing the function of CapChR2 at low Ca²⁺ and physiological Mg²⁺ and K⁺ concentrations and the direct comparison to CatCh (see below).

Detailed comments and suggestions:

1) Serum Mg²⁺ level is about 1 mM and Mg²⁺ concentration of 1-2 mM is often present in the experimental bath solutions (e.g. those for Fig. 1). The authors correctly mention (Line 144-146) "Previous studies on C2-LC-TC revealed that the contribution of Ca²⁺ to the photocurrent is low or even negligible at typical vertebrate ionic conditions (120 mM [NaCl]_e, 2 mM [CaCl₂]_e and 2 mM [MgCl₂]_e). It seems that also for CapChR2, extracellular Mg²⁺ is important because 2 mM Mg²⁺ is reducing Na⁺ currents by 25% (Fig. S10F) and doubles closure time (from 150 to 300 ms, Fig S10I) but data on Ca²⁺ currents are lacking. Because the authors discuss an alternative gating model for Ca²⁺ than for Na⁺ conductance (Fig 8 and discussion) it would be essential that the effect of 1 and 2 mM Mg²⁺ on Ca²⁺ photocurrents and Ca²⁺ imaging (and not only on Na⁺ currents as in Fig. S10) is shown.

2) In my view, Fig. 4 and SI12 are showing the most important Ca²⁺ imaging data on Ca²⁺ entry through CapChR2 at physiological Ca²⁺ concentration of 2 mM but, in contrast to most other figures, direct comparison to CatCh was not performed. Also, this data has been obtained without Mg²⁺ and with only 1 mM K⁺ and with 1mM CsCl and details on the intracellular solution are not given. The use of 1 mM caesium should be justified.

Thus, the authors are encouraged to perform Ca²⁺ imaging with CapChR2 (CapChR1 not required) in physiological solutions (140 mM NaCl, 4-5 K⁺, 2 Ca²⁺, 1-2 Mg²⁺ and without caesium and, importantly, should directly compare these data at least to CatCh (or even better to CrChR2 ET-TC). This can be done in either non-patched cells (Fura2-AM or Cal630-AM) at their physiological resting membrane potential, which might be sufficiently negative if caesium is omitted and without very low K⁺ (1 mM) which is both known to inhibit K⁺ channels. Alternatively, patched cells at -80 mV as in Fig 4 can be used but the intracellular solution should be disclosed and should be as physiological as

possible (e.g. Ca²⁺ clamped at 100 nM by Ca²⁺/Ca²⁺EGTA or by perforated patch clamp to not buffer Ca²⁺).

3) The manuscript contains several of non-harmonized abbreviations which confuses the reader. All abbreviations should be carefully considered if really needed (ACR, CCR, KCR, GECA, DVCF, "0mM [DVC]", sCapChR1), should be fully explained including name of species at first use (CrChR2, CoChR, TsChR, NMGCl) and, importantly, one abbreviation should be used consistently for the same thing (CatCh = CrChR2-L132C = C2-LC; CrChR2 = C2 = ChR2, CoChR=Co, ...). Also, with the different meanings of numbers at the end it is fully confusing that CapChR1 is derived from CrChR2 and CapChR2 is from CoChR. Although synonyms may be fancy, please consider to just use the following abbreviations: CrChR2, CoChR, TsChR or (better if explained correctly) C2, Co, Ts for the protein and SD-LC-TC-NE for the mutation (including position at first use). Thus, CapChR2 could be "Co-SD-LC-TC-NE" (14 instead of 9 letters). With the current non-uniform mixture of abbreviations some of the text and figures are difficult to follow.

4) Although fully clear to the ChRs experts, the rationale of using NMGCl in many experiments as controls and the implications of the results obtained thereof should be better explained to the non-expert reader.

5) To be able to compare light intensities used with other manuscripts, please provide intensity in mW/mm² throughout the manuscript, especially in Figure 4 and S12). Also, light intensity should be mentioned in all legends (e.g. missing in Fig 1-3).

6) Why was the "famous" H134R mutation not included? Although experiments with this mutation are not required for revision, an additional effect of this mutation could be at least be discussed.

7) The effect of Apam, Pax, RyR and NFA in Fig. 6 should be better explained. Maybe add these blockers into the drawing in Fig 6 I?

8) The use of Cal630 is excellent and in this regard the authors could point towards the fact that 380 nm in Fura2 imaging seems to be able to activate the proteins already as can be seen from the action spectra (Fig. S15) and the clear effect at baseline (Fig. S12b).

9) Minor comments:

Reference to the jRCaMP1 should be given on p17.

The legends to supplemental figures should start with "Figure Sx:" and not with "Six:" to match the text (Fig. Sx).

Of note, the “reporting summary” file was not included in the initial submission and thus was not reviewed.

Reviewer #3 (Remarks to the Author):

The authors measured Ca²⁺ permeability of 12 different published CCRs with high cation conductance to choose the best to start with to increase permeability by mutagenesis. CoChR was the most permeable (see Fig 1D) and the authors chose to mutate wild-type CoChR along with the well characterized and most commonly used CCR, CrChR2. The most conductive mutant the authors made was the quadruple mutant CoChR-S43D-L112C-T139C-N238E, which they named a “Calcium permeable ChannelRhodopsin” (CapChR2), and the same four mutations in the less efficient CrChR2, which they examined first, was named CapChR1. They find that the ratio of permeability of Ca²⁺ to that of Na⁺ is 1.9, which is novel since Na⁺ is typically more permeable than Ca²⁺. Another unusual feature is that the permeability of K⁺ is similarly low as that of Na⁺. Finally, to test for usefulness of the CapChRs as optogenetic tools, the authors show that CapChR2 is capable of Ca²⁺ influx higher than that of Na⁺ in mouse hippocampal neuron culture and in a *Drosophila* explant brain preparation.

CRITIQUE:

The authors have made many measurements all expertly performed. I have two major concerns over measurements lacking in the manuscript that bear directly on the utility of the CapChRs and their claimed properties compared to existing optogenetic tools. Both of these major revisions are needed before the manuscript be considered for publication.

1. Fig 1D shows Ca²⁺ permeability of CoChR and other less permeable CCRs by Fura-2 imaging of calcium influx. Since the authors’ report increased permeability by their mutagenesis of these CCRs, most notably CoChR, it is essential to include in Fig 1D at least the data for CapChR1 and CapChR2 recorded in the same conditions as CoChR. Also the presumably much higher permeability will explain why the authors’ 2 quadruple mutated proteins merit the name “Calcium-permeable ChannelRhodopsins”, whereas CoChR is not so named.

Furthermore, in Zhuo-Hua Pan’s study (authors’ reference 43) two enhanced CCRs made from CoChR are reported, one containing 1 mutation (CoChR-L112C) and the other 3 mutations (CoChR-H94E-L112C-K264T). Pan’s two enhanced CCRs increased light-induced current amplitudes by 2-fold and 3-fold over the wild-type posted here. The CapChRs share one mutation with Pan’s mutants and therefore at least a 2-fold increase is expected. It would be important to interpretation of this submission’s data whether the additional mutations in the central gate in CapChRs increased Ca²⁺ permeability significantly over that of CoChR-L112C.

2. The authors’ proposed model for Ca²⁺-permeation and inward rectification in CapChR2 requires that their central gate mutations create a calcium-binding site (Fig. 8). Such a site is suggested by MD simulations, but not by data. The lack of an absorption shift in CapChR2 compared to the parent

CoChR and no change upon Ca²⁺ addition argues against a mutation-induced new Ca²⁺ binding site in the central gate since the gate is near the chromophore. For a thorough analysis, the manuscript needs measurements of the stoichiometry and K_d for Ca²⁺ binding in the parent and mutants. The authors have already obtained the ChRs in purified forms suitable for Isothermal Titration Calorimetry, typically used for this purpose.

All of our point-by-point replies are detailed below. Comments and answers are shown in green.

Reviewer #1 (Remarks to the Author):

Summary:

Aiming for ChR variants with higher calcium selectivity the authors combined mutations, which have been implicated in increasing CrChR2 Ca²⁺-conductance. The comparison of different CrChR2 combinatory mutants by patch-clamp experiments yielded the calcium-permeable CrChR2 quadrupole mutant CapChR1. The transfer of the corresponding set of mutations to CoChR, which the authors selected based on a calcium imaging assay, led to the calcium-permeable ChR variant CapChR2. CapChRs exhibit a high Ca²⁺ conductance at negative voltages. The authors point out utility of CapChRs as genetically encoded calcium actuators in ND7/23 cells, in primary hippocampal neuron cultures, in organotypic hippocampal slices and in M4/6 mushroom body output neurons from *Drosophila* brain explants.

We thank the reviewer for the time taken to critically review our manuscript and the comments made in order to improve it.

Major comments:

p1 line 22, p.8 lines 177-185, Fig. S4

“...the resulting CoChR (Co-SD-LC-TC-NE) expressed well with reliable membrane localization. It exhibited improved calcium permeation at negative holding potentials, similar to C2-SD-LC-TC-NE (Fig. 2D and H). Recordings on Co-SD-LC-TC-NE also revealed a high relative calcium permeation ($I_{Ca}/I_{Na} \sim 1.9$), a positive Erev-shift in calcium buffer in comparison to the WT, and low proton permeability (Fig. 2J and K, S2H). According to these properties, we designated C2-SD-LC-TC-NE and Co-SD-LC-TC-NE as CapChR1 and CapChR2 respectively (Calcium permeable ChannelRhodopsins)...” Fig. S4 shows that the Erev-shift upon replacement of external sodium by calcium and the relative calcium permeation at -80 mV (I_{Ca}/I_{Na}) are likely not significantly different in CapChR1, CapChR2 and ChR2 S63D (Plazzo et al., 2012). The authors included the previously described S63D mutation into their quadruple mutants CapChR1 and CapChR2. Hence, the authors description of CapChRs having “strongly improved Ca²⁺ permeation” (p. 1 line 22) is not sufficiently supported by their results. A potentially beneficial property of CapChR2 is its higher photocurrent density (Fig S4 C), which may result from good plasma membrane expression. Please perform significance tests for the comparison of the photocurrent densities.

The reviewers' comment in general is correct. The stationary I_{Ca}/I_{Na} between S63D, CapChR1 and CapChR2 is similar, but substantially higher than the respective parental wt, under the conditions tested. But, as noted correctly by the reviewer, the current densities of CrChR2 S63D are significantly lower than both CapChR1 and CapChR2 in our additional testing (now Fig. S7, see below). However, we would like to stress that this is not the only beneficial property of the CapChRs: our measurements also provide evidence for a lowered sodium permeation in both peak and stationary photocurrents (e.g., Fig. 3). In this context, it is also worth noting that prior measurements on CrChR2 S63D have only reported an $I_{Ca}/I_{Na} \sim 0.4$ in comparison to the ~ 0.3 of the WT tested in the previous study (Plazzo et al, 2012). Thus, it was a positive surprise to find such an unexpected increase in stationary I_{Ca}/I_{Na} in this mutant under our chosen conditions. More importantly, this effect does not extend to the S63D peak currents, which is still substantially carried by sodium according to our measurements. Here, we refer to our sentence: “*Crucially, sodium permeation was reduced for both peak and stationary photocurrents, even though currents were generally larger*” (p7, line 163-164).

This is one vital improvement made on CapChR1, which shows reduced sodium permeation upon initial illumination. To illustrate and quantify this point, we have added Supplementary Figure 4 (also shown below). Moreover, we would like to bring to your attention that the main advantage of CapChR1 is its reduced sodium permeation, which served as template for the development of CapChR2. The latter is our recommended construct and induces a robust light-activated Ca^{2+} -influx under multiple experimental conditions.

S4: CapChR1 displays suppressed Na^+ -permeation at negative holding potentials. A) Representative photocurrent traces of CrChR2 S63D and CapChR1 at -80 mV in ND7/23 cells under the denoted buffer conditions. B) Stationary (I_{stat}) and peak (I_{peak}) $I_{\text{Ca}}/I_{\text{Na}}$ of the denoted derivatives at -80 mV holding potential (Mean \pm S.E.M.). C) Stationary and peak photocurrent densities of the denoted constructs at -80 mV holding potential (Mean \pm S.E.M.). D) $I_{\text{Ca}}/I_{\text{Na}}$ of the stationary photocurrent for all variants described in previous Chapters (Mean \pm S.E.M). E) Visualization of the advantages provided by CapChR1, including increased $I_{\text{Ca}}/I_{\text{Na}}$ for both peak and stationary photocurrents in conjunction with improved photocurrent densities. Within a mutant: paired, two-sided Wilcoxon-Mann-Whitney-Test; Between mutants: unpaired, two-sided Wilcoxon-Mann-Whitney-Test; * $P \leq 0.05$, ** $P \leq 0.01$, *** $P \leq 0.001$, **** $P \leq 0.0001$.

S7: Electrophysiological characterization of CrChR2, CoChR and PsChR variants. A) $I_{\text{Ca}}/I_{\text{Na}}$ of the stationary photocurrent at pH 7.2 and -80 mV (Mean \pm S.E.M., $N=4-12$, dots represent single measurements). B) Stationary photocurrent density of the denoted variants at 70 mM $[\text{CaCl}_2]_e$, pH 7.2 and -80 mV (Mean \pm S.E.M., $N=4-12$, dots represent single measurements). C) Estimated reversal potentials under different ionic conditions for selected WT ChRs, single mutants and CapChR1 and 2 (Box middle line: Mean, Box edges: \pm S.E.M, Whiskers: \pm SD, $N=4-12$, dots represent single measurements). Two-sided, unpaired Wilcoxon-Mann-Whitney-Test: * $P \leq 0.05$, ** $P \leq 0.01$, *** $P \leq 0.001$, **** $P \leq 0.0001$.

The authors should, as outlined in the subsequent comments, provide further evidence for CapChRs superiority to prior existing genetically-encoded calcium actuators.

Fig. 3 D,E,F, Fig S4 C, p.4 lines 89 to p.5 line 103, p.10 lines 219-226

Measuring the difference of the photocurrent reversal potentials at varying ionic conditions is of key importance for relative cation permeability determination (e.g. P_{Ca}/P_{Na}). The authors state that “For C2-LC, we confirmed large inward photocurrents in presence of 140 mM [NaCl]_e and decrease in amplitude upon replacement of Na⁺ by Ca²⁺ (Figure 3A, D and G), indicating poor conductance for Ca²⁺. This conclusion is also supported by the negative Erev-shift (ΔE_{rev}) upon complete sodium replacement (Fig. 3D). Conversely, both CapChRs exhibit a calcium-dependent increase of inward currents (Fig. 3B and C). At -80 mV, currents increased ~3-fold for CapChR1 (Fig. 3E and H) and ~5-fold for CapChR2 when Na⁺ was replaced with Ca²⁺, coupled with minimal impact on the reversal potentials for CapChR2 (Fig. 3F and I).” These statements are questionable, because the variability of the measured reversal potentials, in particular in the presence of 70 mM calcium, is much higher for CapChR2 (Fig. 3 F) than the variability of the measured reversal potentials for CapChR1 (Fig. 3 E) and C2-LC (Fig. 3 D). A likely reason for the observed variability is voltage drifts in the performed patch-clamp experiments. Therefore additional experiments for CapChR1 and CapChR2 reversal potential determination at 144 mM [NaCl]_e, 5 mM [CaCl₂]_e + 134 mM [NaCl]_e and 70 mM [CaCl₂]_e should be performed (using solution exchange) which include measurements at identical ionic conditions for voltage drift control. The authors could then exclude measurements, which show considerable voltage drifts. The high variability of the measured reversal potentials is also apparent in Fig. S4 C, which compares the calcium permeation of previously described ChR variants. The authors may corroborate their findings by additional measurements as outlined above. Alternatively, the corresponding paragraph (p.4 lines 89 to p.5 line 103) should be carefully rewritten.

We understand the reviewers' concerns regarding the P_{Ca}/P_{Na} determination. However, we would like to point out that we tried to address this peculiarity in p20, lines 452-472. In the case of CrChR2, previous studies have described the inapplicability of the GHK model due to voltage-dependent cation binding. The reason is that Ca²⁺ needs to overcome a substantial energy barrier to permeate through the channel which is only achieved at very negative or positive voltage. The channel is NOT an ohmic resistance for Ca²⁺ in contrast to many mammalian Ca²⁺ channels (Gradmann et al, 2011 and Schneider et al, 2013; PMID: 21889442 and 23823227). This is the reason why we employed the I_{Ca}/I_{Na} at -80 mV as the comparison criteria for Ca²⁺-permeation (e.g., in Fig. 3). We would also like to emphasise that, for example, although the spread in the case of CapChR2 at 70 mM [CaCl₂]_e varies in degree, it leads to a similar mean in the reversal potential under identical conditions (Fig. 2 and Fig. S12F; -0.6 ± 1.9 mV vs. -1.0 ± 1.5 mV respectively; Mean \pm S.E.M). We are therefore quite confident in the reproducibility of these results under our conditions. It is worth mentioning that we performed all current-voltage measurements with solution exchange, at the same setup, under the same experimental conditions and with the same settings. Therefore, any experimental voltage-drift is comparable in all constructs described in this study. We therefore prefer to modify our formulations and have consequently described the results more carefully (p 9-10, lines 220-227).

Figure 1 D, Figure 4, p. 4 lines 77-79 and p. 12 lines 263-266, Figure 5, p. 20 lines 473-478

By employing the Fura-2 and Cal-630 calcium imaging assays, the authors quantify intracellular Ca²⁺ accumulation while the intracellular solution is continuously exchanged against the solution in the patch pipette. Thereby intracellular calcium accumulation not only depends on the calcium permeabilities of the ChR variants but also on e.g. the rate of diffusional exchange between cells and patch pipette, on the cellular surface to volume ratio and on ChR plasma membrane expression. The inset in Figure 1 D should therefore not be labeled with “Ca²⁺-permeability”. Due to the outlined limitations of the calcium imaging assay, it is questionable if sentences like “However, both CoChR

and TsChR showed larger calcium conductance than C2-WT and similar to C2-LC (Figure 1C and 78 D).“ and “In mouse neurons, we observed that CapChR2 triggered a 3-fold stronger Ca²⁺ signal at physiological concentrations compared to C2-LC without substantial differences in decay kinetics (Figure 5).” are accurately describing the results.

Although we understand the hesitancy regarding patch-clamped imaging, we disagree on this point with the reviewer. Figure 1 depicts a calcium-imaging assay with cultured cells without patching (Fura-2-AM loaded ND/23 cells) and in theory should reflect the calcium accumulation by different ChRs under the conditions tested. We have added details to the figure description to make this clearer. According to all the reviewers’ comments, we now have also added such an assay for CrChR2 L132C, CoChR WT, CapChR1 and CapChR2 at physiological ion concentrations (Fig. S16). In both “undisrupted” (Fig. 1 and Fig. 7 and S16) and pipette-accessed imaging experiments (Fig. 4 and 5), CapChR2 stands out as the superior construct, demonstrating consistency in our results across different tissue types and assays. Even if one might assume negative effects of the pipette buffer exchange on calcium imaging experiments, comparing different constructs in the same way should reflect different opsin performances reliably. We however agree that the quantification describing Figure 5 might not entirely reflect the whole picture and therefore have revised associated sentences (p 14 line 309 and p21 line 482).

The authors should also address the question if the blue light pulses, which were applied in the calcium imaging assays evoked saturating or subsaturating photocurrents in the investigated ChR variants (e.g. Fig. 5, photocurrents of C2-LC-YFP and CapChR2-YFP evoked by 50 ms flashes of 470 nm light at an intensity of 1 mW/mm²).

We measured at saturating conditions and have added details to all figure legends.

Minor Comments:

p.2 lines 33-35

“Several natural and engineered ChRs have been shown to be selective for H⁺, Na⁺ or Cl⁻–7...”
The engineered ChRs, which are described in the given references show considerable permeation of other cations in addition. Therefore, changing the sentence to e.g. “...engineered ChRs have been shown to exhibit a higher selectivity for H⁺, Na⁺ or Cl⁻–7...” would provide a more precise description.

Good suggestion, we now used the proposed sentence.

p.4 lines 88-89

“However, in most of these cases, Na⁺ vs. Ca²⁺ permeation has not been rigorously examined.”
Please clarify by referencing for which of the reexamined ChR variants Na⁺ vs. Ca²⁺ permeation has been examined before and for which of the reexamined ChR variants Na⁺ vs. Ca²⁺ permeation has not been examined before.

We have rephrased the sentence to better reflect what we intended to communicate to the reader, which is: the examination of other studies was not uniform, and under strongly varying conditions (p4 lines 90-91).

p. 5 lines 118-123, p. 7 lines 151-153, p. 19 line 452 – p. 20 line 456, p. 18 line 401 – p. 19 line 433, Fig. 8

“Varying E90 mutations modified either the Na⁺/H⁺ ratio or even converted C2 into an anion

channel^{4,38}. Similarly, homotetrameric Na⁺-selective channels have been converted into Ca²⁺-channels by introduction of negative charges to their selectivity filters^{39,40...}

“Based on our initial voltage-clamp experiments (Fig. S3), and following the logic outlined above, additional negative charge in the proximity of E90 might further increase relative Ca²⁺ permeation.”

“... the aforementioned voltage-dependent barrier for the Ca²⁺ conductance can be massively reduced by modification of the CrChR2 central gate residues, primarily S63D and N238E in CapChR1, which are similar to selectivity filters in other calcium channels (Fig. S14) ³⁹.”

The authors repeatedly mention similarities between the modified central gate in CrChR2 mutants and selectivity filters of calcium channels. That interesting hypothesis may eventually be supported by a supplementary figure showing a high-resolution structure based comparison of the calcium channel selectivity filter and the central gate of CrChR2. The MD simulation provides limited insights in that regard, because it is based on the closed channel structure and does not directly address the potential role of the CapChR mutations in calcium permeation.

That is an interesting suggestion. Accordingly, we have added a figure to illustrate our point (Fig. S5). The carboxylic residues at the central gate could putatively assist in calcium permeation and attraction at the central gate. However, it is worth noting that the study on the conversion of Na_vAb to Ca_vAb was done with a tetrameric channel with a highly symmetric pore (Tang et al 2014, PMID: 24270805; Fig. S5). The most interesting aspect in this context is therefore the conversion of a sodium channel into a calcium channel via introduction of carboxylic acids at the selectivity filter, as outlined in the manuscript (p5, lines 119-127).

S5: Structure of homotetrameric Ca_vAb (PDB: 4MVQ) and monomeric CrChR2 and derivative CapChR1. A) Top view of Ca_vAb tetramer with voltage sensing domains (VSD) and pore domains with zoom-in (right) on bound calcium (rose). B) Side-view of the selectivity motif of Ca_vAb, where carboxylic aspartates are mainly responsible for initial calcium binding and selectivity. In Ca_vAb (parental protein is Na_vAb, a sodium channel), E177, S178 and M181 were substituted with Asp to switch selectivity to Ca²⁺. C) Side-view of in silico equilibrated CrChR2 WT and its derivative D) CapChR1 (based on PDB-ID 6E1D) with cofactor all-trans-retinal bound and side chains of amino acids possibly involved in cation binding and selectivity highlighted as licorice. The water-filled pores are shown as blue surfaces. Cation uptake on the extracellular side might be mediated by carboxylic residues E90, E97, E101 and D253, while Ca²⁺ selectivity is increased by introducing S63D and N258E at the central gate as additional negative charges (highlighted in zoom-ins as side- and top-views, mutated residues are colored).

Figure S7, Fig. S15, p. 8 line 186-19

CapChR2 shows a 4 nm red-shift and band narrowing at high $[CaCl_2]_e$, a possible indicator for a Ca^{2+} -binding site close to the protonated retinal Schiff base (RSBH⁺) (Fig. S7). To analyse the impact of the mutations on the spectral properties of CoChR and CapChR2, we recorded absorption spectra from purified proteins (Fig. S15). We observed no absorption shift in CapChR2 compared to the parental WT and no change upon Ca^{2+} supplementation. We presume that the small spectral changes seen in the action spectra upon replacement of Na^+ with Ca^{2+} might only be present at negative membrane voltages.”

Please prove the significance of the observed 4 nm red-shift (Fig. S7), in particular as this finding is not in line with the recorded absorption spectra. The presumption that spectral changes are voltage dependent should be corroborated by measurements of action spectra at different membrane potentials. Alternatively, the corresponding paragraph may be deleted.

In line with our argumentation, the minimal shift was observed in the action spectra recorded at negative voltages, which is not necessarily seen in absorption spectra in detergent. However, since the action spectra have not been measured many times (with statistically insignificant shifts) and this issue is not really relevant for our main conclusions, we agree to tune down our argument in the corresponding paragraph to only one sentence (p8, lines 193-195).

Figure 6, p. 15 lines 347-350, p. 16 lines 359-361

“...under our recording conditions, the 10-fold chloride gradient across the plasma membrane should result in a Nernst potential of about -60 mV. The CACl-channels that allow this secondary inward current (chloride efflux) are potentially distant from the cell soma and thus possibly see a reduced chloride gradient⁴⁹.”

“In current clamp measurements, depolarization due to CapChR2 activation led to robust spiking (Fig. 6F-H), but not in the presence of NFA, confirming that the photocurrent of CapChR2 alone is not enough to drive neuronal spiking.”

The finding that even though the Nernst potential for chloride was about -60 mV, CACl-channel activation by CapChR2 led to depolarizing currents, which were essential to drive neuronal spiking, is explained by the localization of CACl channels distant from the soma, where local chloride gradients shift the Nernst potential to more positive voltages. Please provide a reference that shows that CACl-channels are expressed predominantly distant to the soma in hippocampal neurons.

Figure 7, p. 17 lines 377-392

Thank you for this suggestion. Unfortunately, direct descriptions of subcellular localization are not available. However, two observations support our speculation. There is evidence suggesting CaCCs are preferentially expressed close to NMDA receptors (Huang et al 2012; PMID: 22500639). Thus, this would suggest they are distant from the soma, closer to dendrites. In addition, the impact of varying chloride gradients across neurons has been reported in the context of optogenetics, where anion conducting channelrhodopsins have been shown to elicit spiking instead of inhibition in the axon due to changed chloride concentrations (see Rost et al. 2022; PMID: 35835882), which would lend credence to our interpretation of results.

Overall, because we did not address this question directly, we now acknowledge that our idea about a polarised distribution of CaCCs is a speculation, give a reference to the characteristic slow tail current that was reported to be due to CaCCs, and note that more work would be needed to address the subcellular distribution of CaCCs.

Based on experiments in *Drosophila* M4/6 mushroom body output neurons using jRCaMP1a, a GECI that is used for neural activity imaging, the authors conclude that “These data confirm the specific action of CapChR2 to raise calcium in targeted cells types, in an intact brain.”. Please discuss the

contribution of voltage activated calcium channels on the one hand and the contribution of CapChR2 on the other hand to the observed increase of the intracellular calcium concentration in the *Drosophila* M4/6 mushroom body output neurons.

In principle, the activation of voltage-gated calcium channels should be observable in the case of CapChR1 due to comparable current size and associated depolarisation. Therefore, the difference between CapChR1 and CapChR2 in *Drosophila* experiments should be a reliable reflection of CapChR2-induced increase in cytosolic Ca^{2+} . One thing to note however, is the possibility of calcium induced calcium release from intracellular stores. This would indicate that CapChR2 is an efficient initiator of calcium signalling. This is now discussed in the manuscript (p22 lines 499-504).

Significance:

ChR variants, which show a strongly increased Ca^{2+} -selectivity are an important extension of the optogenetic toolbox. Remote control of intracellular Ca^{2+} concentration by a light-activated genetically encoded Ca^{2+} actuator is of great interest for cell biology, because a large number of cell biological processes depend on Ca^{2+} . CapChRs will potentially prove useful for the investigation of Ca^{2+} signaling processes.

Reviewer #2 (Remarks to the Author):

The manuscript "Channelrhodopsin-based calcium actuators for the photocontrol of calcium signalling" by Fernandez Lahore et al. is a very important work on the rational design of novel Channelrhodopsins (ChRs) that allow selective conducting of Ca^{2+} ions by illumination. In fact, researchers from various disciplines applying optogenetic tools including myself are awaiting such ChR2 desperately because Ca^{2+} is the most important intracellular second messenger involved in all cells of the body. Until to date, optogenetic tools for direct influx of Ca^{2+} from the extracellular space physiologically containing ~ 2 mM Ca^{2+} are lacking and the best available tool CatCh has been characterised at extracellular non-physiologic levels of 70 mM Ca^{2+} .

In the present manuscript, the authors initially recapitulate previous Ca^{2+} imaging data with very high (70 mM) external Ca^{2+} and additionally show that wild-type ChR from other species (CoChR, TsChR) are similarly Ca^{2+} conducting than CatCh, which is derived from CrChR2 (Fig. 1). Subsequently, the authors nicely explain the rational design of a better Ca^{2+} permeable ChR by combining ChRs from different species with previously known mutations promoting Ca^{2+} entry and ChRs kinetics (LC, TC) with novel mutations (NE, SD) at the inner central gate (Fig. 2 and many supplemental Figures towards this goal). This resulted in ChRs with quadruple mutations termed "Calcium permeable ChannelRhodopsins" (CapChR1 and 2, but clearness of abbreviations is discussed below).

CapChR1 and 2 were analysed in depth by electrophysiology (Fig. 3) and, in contrast to CatCh, showed larger currents in 70 mM Ca^{2+} than in 140 mM Na^{+} , indicative of Ca^{2+} over Na^{+} selectivity. Because of the surprisingly strong inward rectification allowed Ca^{2+} currents only at very negative membrane potentials (-60 and -80 mV) Ca^{2+} imaging was performed while patch clamping cells at -80 mV. In these key experiments CapChR2 showed large Ca^{2+} influx even at physiological Ca^{2+} concentrations of 2 mM (Fig. 4) but unfortunately Ca^{2+} imaging experiments in the presence of Mg^{2+} or direct comparison to CatCh was not performed. Finally, the authors nicely applied CapChRs in neurons (Fig. 5), organotypic hippocampal slices (Fig. 6) and *Drosophila* brain preparations (Fig. 7) to show that CapChR2 outperforms CatCh. Finally, using MD simulations the authors try to explain Ca^{2+} conductance and the rectification in CapChRs with a novel mechanism of ChRs gating (Fig. 8).

The results are original, important, timely and of high significance to the optogenetic field and related fields applying optogenetics. The conclusions are based on the extensive data, the methodology is sound with sufficient details explained (with an exception on intracellular solution, see below) and the work meet the expected standards in the field.

I highly recommend publication with minor revision and some more Ca²⁺ imaging experiments showing the function of CapChR2 at low Ca²⁺ and physiological Mg²⁺ and K⁺ concentrations and the direct comparison to CatCh (see below).

We are very thankful to Philipp Sasse for his positive outlook and encouraging comments regarding improvements that can be made to our manuscript.

Detailed comments and suggestions:

1) Serum Mg²⁺ level is about 1 mM and Mg²⁺ concentration of 1-2 mM is often present in the experimental bath solutions (e.g. those for Fig. 1). The authors correctly mention (Line 144-146) "Previous studies on C2-LC-TC revealed that the contribution of Ca²⁺ to the photocurrent is low or even negligible at typical vertebrate ionic conditions (120 mM [NaCl]_e, 2 mM [CaCl₂]_e and 2 mM [MgCl₂]_e). It seems that also for CapChR2, extracellular Mg²⁺ is important because 2 mM Mg²⁺ is reducing Na⁺ currents by 25% (Fig. S10F) and doubles closure time (from 150 to 300 ms, Fig S10I) but data on Ca²⁺ currents are lacking. Because the authors discuss an alternative gating model for Ca²⁺ than for Na⁺ conductance (Fig 8 and discussion) it would be essential that the effect of 1 and 2 mM Mg²⁺ on Ca²⁺ photocurrents and Ca²⁺ imaging (and not only on Na⁺ currents as in Fig. S10) is shown.

This is an interesting point. Our combined datasets and analysis would lead us to interpret that this reduction of the currents in the presence of Mg²⁺ is a reduction of the sodium ion conductance, not calcium ions. We therefore performed calcium imaging under the conditions suggested by the reviewer in the presence and absence of Mg²⁺ (Fig. S16, see below). Interestingly, this could be the case according to our new data: for both CapChRs, the mean response is independent of the extracellular Mg²⁺, indicating that even at low concentrations of Ca²⁺, the former does not inhibit the influx of the latter. In CrChR2 L132C, there is a slight increase in response (not statistically significant) once Mg²⁺ is removed from the buffer. It might also be noteworthy that in the algae of origin, with a -170 mV resting membrane voltage, inhibition of photocurrents by Mg²⁺ are even more significant (Holland et al. 1996, PMID: 8789109). This supports the claim that the inhibition is voltage dependent.

2) In my view, Fig. 4 and S112 are showing the most important Ca²⁺ imaging data on Ca²⁺ entry through CapChR2 at physiological Ca²⁺ concentration of 2 mM but, in contrast to most other figures, direct comparison to CatCh was not performed. Also, this data has been obtained without Mg²⁺ and with only 1 mM K⁺ and with 1mM CsCl and details on the intracellular solution are not given. The use of 1 mM caesium should be justified. Thus, the authors are encouraged to perform Ca²⁺ imaging with CapChR2 (CapChR1 not required) in physiological solutions (140 mM NaCl, 4-5 K⁺, 2 Ca²⁺, 1-2 Mg²⁺ and without caesium and, importantly, should directly compare these data at least to CatCh (or even better to CrChR2 ET-TC). This can be done in either non-patched cells (Fura2-AM or Cal630-AM) at their physiological resting membrane potential, which might be sufficiently negative if caesium is omitted and without very low K⁺ (1 mM) which is both known to inhibit K⁺ channels. Alternatively, patched cells at -80 mV as in Fig 4 can be used but the intracellular solution should be disclosed and should be as physiological as possible (e.g. Ca²⁺ clamped at 100 nM by Ca²⁺/Ca²⁺EGTA or by perforated patch clamp to not buffer Ca²⁺).

Cs⁺ is commonly used as a blocker of K⁺-channels in patch-clamp experiments in our laboratory for more stable patches. We could not ascertain any detrimental effects of Ca²⁺ on characterization in the past. See usage of Cs⁺ in buffers for previous ChR characterization in: Wietek et al. 2014; Vierock et al. 2017 and Oppermann et al. 2019 (PMID: 24674867, 28855540 and 31346176 respectively).

We have performed Fura-2-AM imaging under the conditions suggested by the reviewer (Fig. S16). We used C2-LC since the smaller currents of CrChR2 ET-TC will expectedly lead to reduced calcium permeation. We found the performance of CrChR2 L132C, CoChR and CapChR1 to be comparable to each other. Here, we would like to draw attention to the fact that CapChR1 exhibits reduced sodium permeation (Fig. 2 and 3, S2 and 3), so the same calcium permeation is presumably induced by smaller current amplitudes. Notwithstanding, our recommended construct CapChR2 elicits a ~5-fold greater response than CrChR2 L132C in this experiment, despite the reduced hyperpolarization of the cell type used, and calcium influx is independent of the extracellular Mg²⁺ concentration. Therefore, these results support our prior conclusions defining CapChR2 as superior to CapChR1, CrChR2 L132C and CoChR.

S16: Fura-2 AM imaging on C2-LC (red), CoChR WT (blue), CapChR1 (green) and CapChR2 (orange). A) (top) Overview of experimental setup: ND7/23 cells with a resting membrane potential were illuminated with 470 nm (~0.08 mW/mm², saturating light intensities) light to allow influx of calcium ions through the expressed ChR in the presence and absence of bath Mg²⁺. (bottom) Mean imaging response of the denoted constructs under the two measured conditions (Mean as colored line and single measurements in light grey). B) Quantified peak responses after 10 s of illumination. Each dot represents one cell (Mean ± S.E.M. ; N = 21-32). Two-sided, unpaired Wilcoxon-Mann-Whitney-Test: *P ≤ 0.05, **P ≤ 0.01, ***P ≤ 0.001, ****P ≤ 0.0001.

3) The manuscript contains several of non-harmonized abbreviations which confuses the reader. All abbreviations should be carefully considered if really needed (ACR, CCR, KCR, GECA, DVCF, "0mM [DVC]", sCapChR1), should be fully explained including name of species at first use (CrChR2, CoChR, TsChR, NMGCl) and, importantly, one abbreviation should be used consistently for the same thing (CatCh = CrChR2-L132C = C2-LC; CrChR2 = C2 = ChR2, CoChR=Co, ...). Also, with the different meanings of numbers at the end it is fully confusing that CapChR1 is derived from CrChR2 and CapChR2 is from CoChR. Although synonyms may be fancy, please consider to just use the following abbreviations:

CrChR2, CoChR, TsChR or (better if explained correctly) C2, Co, Ts for the protein and SD-LC-TC-NE for the mutation (including position at first use). Thus, CapChR2 could be “Co-SD-LC-TC-NE” (14 instead of 9 letters). With the current non-uniform mixture of abbreviations some of the text and figures are difficult to follow.

We understand the abbreviations might lead to confusion in certain situations; therefore, we have taken care that abbreviations are kept uniform and properly denoted when first mentioned. We have also avoided abbreviations if not necessary. Additionally, where we deemed helpful in figure legends, we have added full name and abbreviation to avoid reader confusion.

We would like to keep the names CapChR1 and CapChR2 due to the intuitive assumption that the second version is better than the first (e.g. as in GCaMP2 over 1). Reading of the sentence: “According to these properties, we designated C2-SD-LC-TC-NE and Co-SD-LC-TC-NE as CapChR1 and CapChR2 respectively (Calcium permeable ChannelRhodopsins)...” should in essence communicate to the reader effectively what the CapChRs are, especially in the context of Figure 2, where it is also clearly stated. CapChR1 and CapChR2 were chosen so that non-expert readers intuitively will know that CapChR2 is the better version. Further, we have carefully looked at the abbreviations to see if they are uniformly used.

4) Although fully clear to the ChRs experts, the rationale of using NMGCl in many experiments as controls and the implications of the results obtained thereof should be better explained to the non-expert reader.

NMGCl is commonly used to evaluate the proton permeation in electrophysiological experiments. Due to the large size of NMG^+ , in a buffer with NMGCl and at pH 7.2 the osmolarity is preserved and the dominant permeated ions are protons. Simultaneously, replacing Na^+ with NMG^+ preserves the charge balance across the membrane, allowing for assessment of ion permeation without disturbing the electrochemical gradient. We added a brief sentence to explain on p 4, line 94-95.

5) To be able to compare light intensities used with other manuscripts, please provide intensity in mW/mm^2 throughout the manuscript, especially in Figure 4 and S12). Also, light intensity should be mentioned in all legends (e.g. missing in Fig 1-3).

Added.

6) Why was the “famous” H134R mutation not included? Although experiments with this mutation are not required for revision, an additional effect of this mutation could be at least be discussed.

This is an interesting mutation because, in contrast to the L132 position, H134 is part of the conducting pore. The H134R mutation improved the Na^+/H^+ ratio and reduced desensitisation (Berndt et al 2011, PMID: 21504945; <https://edoc.hu-berlin.de/handle/18452/17002>) but has never been associated with an improved calcium permeation/selectivity. To improve current size, we used the T159C mutation, which also increases retinal binding and is in another helix, further away (Berndt et al 2011, PMID: 21504945; Ullrich et al. 2013, PMID: 23134970).

We were also hesitant to include a positive charge at the inner gate of the channel, which is proposed to be a site for Na^+ accumulation (see Schneider, Grimm and Hegemann, 2015). In fact, experiments combining CapChR1 with the similar H134K replacement had a negative impact on channel function. Further, even if the mutation generally enhances current size at high Na^+ , we expect detrimental effects on divalent cation permeation due to electrostatic repulsion or reduced binding.

7) The effect of Apam, Pax, RyR and NFA in Fig. 6 should be better explained. Maybe add these blockers into the drawing in Fig 6 I?

The effects have been included in Fig 6I and depicted accordingly.

8) The use of Cal630 is excellent and in this regard the authors could point towards the fact that 380 nm in Fura2 imaging seems to be able to activate the proteins already as can be seen from the action spectra (Fig. S15) and the clear effect at baseline (Fig. S12b).

This is correct, the monitoring light indeed already also excites CapChRs due to absorption in the UV. We added a line to include this observation (p 14 lines 303-304).

9) Minor comments:

Reference to the jRCaMP1 should be given on p17.

The legends to supplemental figures should start with "Figure Sx:" and not with "Six:" to match the text (Fig. Sx).

Of note, the "reporting summary" file was not included in the initial submission and thus was not reviewed.

Of course, it is included now.

Review signed by Philipp Sasse, Bonn

Reviewer #3 (Remarks to the Author):

The authors measured Ca²⁺ permeability of 12 different published CCRs with high cation conductance to choose the best to start with to increase permeability by mutagenesis. CoChR was the most permeable (see Fig 1D) and the authors chose to mutate wild-type CoChR along with the well characterized and most commonly used CCR, CrChR2. The most conductive mutant the authors made was the quadruple mutant CoChR-S43D-L112C-T139C-N238E, which they named a "Calcium permeable ChannelRhodopsin" (CapChR2), and the same four mutations in the less efficient CrChR2, which they examined first, was named CapChR1. They find that the ratio of permeability of Ca²⁺ to that of Na⁺ is 1.9, which is novel since Na⁺ is typically more permeable than Ca²⁺. Another unusual feature is that the permeability of K⁺ is similarly low as that of Na⁺. Finally, to test for usefulness of the CapChRs as optogenetic tools, the authors show that CapChR2 is capable of Ca²⁺ influx higher than that of Na⁺ in mouse hippocampal neuron culture and in a Drosophila explant brain preparation.

We thank the reviewer for the careful assessment of the manuscript and the feedback provided.

CRITIQUE:

The authors have made many measurements all expertly performed. I have two major concerns over measurements lacking in the manuscript that bear directly on the utility of the CapChRs and their claimed properties compared to existing optogenetic tools. Both of these major revisions are needed before the manuscript be considered for publication.

1. Fig 1D shows Ca²⁺ permeability of CoChR and other less permeable CCRs by Fura-2 imaging of calcium influx. Since the authors' report increased permeability by their mutagenesis of these CCRs, most notably CoChR, it is essential to include in Fig 1D at least the data for CapChR1 and CapChR2 recorded in the same conditions as CoChR. Also the presumably much higher permeability will explain why the authors' 2 quadruple mutated proteins merit the name "Calcium-permeable ChannelRhodopsins", whereas CoChR is not so named.

We attempted to measure under the conditions suggested by the reviewer. However, the data acquired was uninterpretable. This was caused by the fact that, as mentioned by reviewer #2, Fura-2 fluorescence acquisition already increased the signal at high concentrations of calcium (the condition of 70 mM used in Figure 1 is quite high) for CapChRs, especially CapChR2 (see Fig. S15). Therefore, we performed a comparison under low concentrations of calcium to have a direct comparison of C2-LC, CoChR, CapChR1 and CapChR2 (Fig. S16), which is in fact more relevant than our screening conditions at high Ca²⁺. The superiority of CapChR2 is directly evident in this assay. However, the advantage of CapChR1 is seen better in electrophysiological analysis, where the reduced sodium permeation in comparison to CoChR is observable (Figure S2). While the latter conducts calcium ions effectively at 70 mM, the conductance for sodium is much higher, and is expected to be the dominantly conducted ion at high extracellular NaCl (Fig. S2).

Furthermore, in Zhuo-Hua Pan's study (authors' reference 43) two enhanced CCRs made from CoChR are reported, one containing 1 mutation (CoChR-L112C) and the other 3 mutations (CoChR-H94E-L112C-K264T). Pan's two enhanced CCRs increased light-induced current amplitudes by 2-fold and 3-fold over the wild-type posted here. The CapChRs share one mutation with Pan's mutants and therefore at least a 2-fold increase is expected. It would be important to interpretation of this submission's data whether the additional mutations in the central gate in CapChRs increased Ca²⁺ permeability significantly over that of CoChR-L112C.

This is an interesting point raised by the reviewer. We now more thoroughly analysed the single mutations introduced in CapChR2 as suggested, with the addition of the double mutant L112C T139C, to assess the impact of each mutation on the Ca²⁺ permeability. We chose to use the same buffers used for the characterization of CrChR2 mutants for better comparison. Although S43D expressed poorly, we were able to measure the I_{Ca}/I_{Na} . As evidenced by our new data (Fig. S6), none of the single or double CoChR mutants exhibit such a pronounced I_{Ca}/I_{Na} . Therefore, it seems to be the case that for CoChR, the combination of S43D, L132C, T139C and N238E is what enables the improved calcium ion permeation, even though residue locations are very similar to C2 according to our model (new Fig. S6 A). Since CoChR L112C T139C displayed similar properties to the single mutant CoChR L112C, we can attribute the enhanced calcium permeation to the addition of S43D and N238E at the central gate.

S6: Calcium conductance of CoChR WT, L112C, L112C T139C, S43D and N238E. A) CoChR homology model depicting the location of the S43, L112, T139 and N238E residues in Helix 1, 3, 4 and 7 respectively (zoomed inlet, based on structure data, PDB ID: 6EID). Grey mesh represents water accessibility according to MD simulations. Dark grey line represents the putative permeation pathway. B) Representative photocurrent traces of WT (dark blue), L112C (sky blue), and L112C T139C (red) and N238E (pale green) recorded from -80 to +40 mV in 20 mV steps in ND7/23 cells under the denoted buffer conditions (blue bar: illumination with saturating, 470 nm light; ~1.9 mW/mm²). C) Stationary I_{Ca}/I_{Na} of the denoted derivatives at -80 mV holding potential. D) I(E) relationships for the

various mutants under varying extracellular conditions (shadows represent the S.E.M; normalised to high [NaCl] at -80 mV).

2. The authors' proposed model for Ca²⁺-permeation and inward rectification in CapChR2 requires that their central gate mutations create a calcium-binding site (Fig. 8). Such a site is suggested by MD simulations, but not by data. The lack of an absorption shift in CapChR2 compared to the parent CoChR and no change upon Ca²⁺ addition argues against a mutation-induced new Ca²⁺ binding site in the central gate since the gate is near the chromophore. For a thorough analysis, the manuscript needs measurements of the stoichiometry and K_d for Ca²⁺ binding in the parent and mutants. The authors have already obtained the ChRs in purified forms suitable for Isothermal Titration Calorimetry, typically used for this purpose.

Here, we disagree with the reviewer that we have no data indicating that there is an effect of the central gate mutants on calcium binding/affinity at the central gate. In fact, the opposite is the case: we performed MD simulations to have theoretical insight into the measured calcium conductance of CapChR2, which is evident and inward-rectified in multiple experiments in the manuscript. In CapChR2 specifically, the introduction of both S43D and N238E increases the I_{Ca}/I_{Na} from ~0.3 in

CoChR L112C T139C to ~ 1.9 in CapChR2 (Fig. S6 C). The lack of an absorption shift in the purified protein does not exclude calcium binding at the central gate upon illumination and negative voltage, which might be a very short-lived state that has little impact on retinal absorption. Thus, a change in ion selectivity and permeation does not necessarily have to be coupled to a spectral shift: in the case of C1C2, neutralisation of the central gate residue E90S in iC⁺⁺, which conducts anions instead of cations, does not lead to a clear absorption shift in the purified sample (~ 480 nm at pH 7, Kato et al. 2012 and Kato et al. 2018; PMID: 22266941 and 30158697) or a shift in action spectra for iC1C2 (Berndt et al 2014, PMID: 24763591). In contrast, replacement of the same residue by a positive charge E90R resulted in a ~ 20 nm red shift of the action spectrum in iChloC (Wietek et al. 2017, PMID: 29097684). Therefore, a shift is possible, but not always present upon change of charge distribution at the central gate and associated ion selectivity changes.

Further, our MD-model represents the closed-state and is meant to give a basis for informed discussion on possible inward-rectification mechanisms of CapChR2. Thus, it was kept in the discussion section of the manuscript. There is no evidence to suggest that calcium binding takes place in the dark.

Although we do understand the motivation behind attributing a certain value of K_d to be able to quantify the improvements in CapChR2, Isothermal Titration Calorimetry is beyond the scope of this manuscript and our present capabilities. We believe this might be interesting in future work but would not provide further insight into the calcium permeation through CapChR2, which we have characterised extensively under several experimental conditions. Further, this method would require voltage application to the sample due to the voltage-dependence of CapChR2, which would require an even more customised setup not available to us at the time.

On the suggestion of reviewer #1, and the reasoning outlined above, we have significantly shortened our paragraph on the action/absorption spectra (p8 lines 193-195).

REVIEWER COMMENTS

Reviewer #1 (Remarks to the Author):

The authors have addressed most of the concerns. I have only two issues left.

A) The Fura-2-AM assay enables the quantification of the calcium accumulation by different ChRs under the conditions tested. The calcium accumulation is not only reflecting the "Ca²⁺-permeability" of the investigated ChRs. It rather is an estimate for the calcium flux, which e.g. also depends on ChR plasma membrane expression. Please label Fig. 1D accordingly.

B) "The resulting C2-SD-LC-TC-NE mutant had enhanced inward Ca²⁺ permeation with strong inward rectification (Fig. 2D and E), similar to that of the single mutant C2-SD (Fig. S3G). Crucially, sodium permeation was reduced for both peak and stationary photocurrents, even though currents were generally larger (Fig. S4)." (p. 7, lines 161-164).

Fig S4 shows that I_{Ca}/I_{Na} of CRChR2 S63D is not significantly different to I_{Ca}/I_{Na} of CapChR1 for stationary photocurrents, whereas the difference becomes evident for peak photocurrents. For clarity, that finding should be described in the main text of the manuscript. Please determine the CapChR1 peak recovery time, which may help to discuss the contribution of stationary photocurrents and peak photocurrents to the results obtained in the different cell types. Possibly the main advantage of CapChR1 compared to CRChR2 S63D under the experimental conditions is not a higher calcium permeability, but a higher stationary photocurrent density, which in consequence leads to a higher calcium influx.

Reviewer #2 (Remarks to the Author):

The authors have adequately replied to my comments and the new data provided is convincing. I recommend publication with three minor requests that are easy to follow:

1) In my view the new supplementary Figure S16 is the most important data in the whole manuscript because it uses non-patched cells and physiological external solution without K⁺ channel block (presumably leading to physiological resting membrane potentials). This is also acknowledged by the authors in their rebuttal "direct comparison of C2-LC, CoChR, CapChR1 and CapChR2 (Fig. S16), which is in fact more relevant than our screening conditions at high Ca²⁺". Along with this notion is mandatory to include Fig. S16 in the main manuscript as additional figure (e.g. after Fig. 4). If word count is an issue is hereby requested to the editor to allow this additional figure and legend. Alternatively, Fig. 4 can be replaced by Fig. S16 because the former it is of less importance showing the same conclusion (but statistics are lacking) as Figure S16 (with statistics) but required patch-clamp induced hyperpolarization, most likely because K⁺ channels are blocked. Please use the same analysis as in Fig 4 (Initial Delta R) for Fig. S16 (currently unclear with "R after illumination") to be able to compare both figures (and the need for hyperpolarization).

For Fig. S16 (and all other figures and legends) please keep abbreviations constant and only use CatCh (as in Fig 1) and not CrChR2-L132C or C2-LC.

2) The reviewer has commented on the confusing abbreviations and de-novo generation of synonyms and the authors have explained to indicate superior function by indices (example of GCaMP given which by different groups consistently with increasing suffixes). Along this idea it is requested to use CatCh for C2-LC (harmonization as requested still not done), CatCh2 for CapChR1 and CatCh3 for CapChR2. It is just confusing and there is no rationale to introduce the

new abbreviation "Calcium permeable ChannelRhodopsins"(CapChR) for a "calcium translocating channelrhodopsin" (CatCh) because "permeable" and "translocating" has essentially the same meaning. By following this request, it will be mandatory for future studies to use CatCh4 and so on for better versions. This would also be best for highlighting the superior function of the new proteins over CatCh.

3) As previously requested, please provide details on the intracellular solution in sections "Whole-cell patch-clamp recordings in ND7/23 cells" and clearly describe if and how intracellular buffering of Ca²⁺ was adjusted.

Review signed by Philipp Sasse

Reviewer #3 (Remarks to the Author):

The authors' principal objective was to combine mutations that are known to increase Ca²⁺ permeability in several light-activated ChRs to produce a multi-mutated ChR form with greater permeability. Success was claimed by producing quadruple mutants they named CapChRs with "strongly improved Ca²⁺ permeation". My main concern was that the authors needed to directly compare the Ca²⁺ permeabilities of the CapChRs with the single-mutant ChRs, suggesting in the same conditions as they used to select the best single-mutant ChRs. The authors attempted to make this measurement. However, they report the data acquired were "uninterpretable" (not exceeding noise levels?) under the ionic conditions that were used with success for the single mutants. They changed ionic conditions reducing calcium concentration and found that CapChR1 is not significantly more permeable than the single mutants, but CapChR2 produced ~3-fold higher currents in those conditions. These results improve the manuscript by introducing a direct comparison, but the results are mixed. CapChR2 shows increased permeability, whereas the necessity to change ionic conditions to make the comparison suggests that the single mutants may have larger currents in some conditions and CapChRs in others.

A less major concern I had mentioned is that the authors state as a fact that the mutations near the central gate introduce a new Ca²⁺ binding site (Page 21, line 468: "... in conjunction with the new calcium-binding site at the central gate."). I believe their proposed site may be correct, but it is not proven, and it would therefore be more accurate to refer to it as "the calcium-binding site proposed in our model". In the authors' response they disagree seemingly based on the assumption that because the two mutations increase the Ca/Na current ratio, there must be a Ca²⁺ binding site. I suggest reading an article from Park and MacKinnon describing principles favoring the observation that weakening of an ion binding site may enhance permeability (Park E, MacKinnon R. 2018. Structure of the CLC-1 chloride channel from Homo sapiens. *eLife* 7:e36629. PMID: 29809153). I do understand that the ITC measurements I had suggested can reasonably be regarded as beyond the scope of this paper.

Minor point: The authors refer to 2 papers, one in BioRxiv, reporting discovery of ChRs with high K⁺ selectivity (KCRs). To be complete, there is a third article that reports more KCRs posted in BioRxiv (doi: <https://doi.org/10.1101/2022.09.26.509509>).

All of our point-by-point replies are detailed below. Comments and answers are shown in green.

Reviewer #1 (Remarks to the Author):

The authors have addressed most of the concerns. I have only two issues left.

A) The Fura-2-AM assay enables the quantification of the calcium accumulation by different ChRs under the conditions tested. The calcium accumulation is not only reflecting the “Ca²⁺-permeability” of the investigated ChRs. It rather is an estimate for the calcium flux, which e.g. also depends on ChR plasma membrane expression. Please label Fig. 1D accordingly.

This is correct. It is a combination of conductance and expression; we have changed the labelling accordingly.

B) “The resulting C2-SD-LC-TC-NE mutant had enhanced inward Ca²⁺ permeation with strong inward rectification (Fig. 2D and E), similar to that of the single mutant C2-SD (Fig. S3G). Crucially, sodium permeation was reduced for both peak and stationary photocurrents, even though currents were generally larger (Fig. S4).” (p. 7, lines 161-164).

Fig S4 shows that I_{Ca}/I_{Na} of CRChR2 S63D is not significantly different to I_{Ca}/I_{Na} of CapChR1 for stationary photocurrents, whereas the difference becomes evident for peak photocurrents. For clarity, that finding should be described in the main text of the manuscript.

We agree with the reviewer that this is worth highlighting further in the main text. We have added sentences to clarify in the main text (p7, lines 163-166).

Please determine the CapChR1 peak recovery time, which may help to discuss the contribution of stationary photocurrents and peak photocurrents to the results obtained in the different cell types. Possibly the main advantage of CapChR1 compared to CRChR2 S63D under the experimental conditions is not a higher calcium permeability, but a higher stationary photocurrent density, which in consequence leads to a higher calcium influx.

The current inactivation as well as the recovery of CapChR1 is a complicated business: both are light-intensity dependent, but additionally depend on the ion concentration (especially Ca²⁺ as seen in Fig. S4A) and on the membrane voltage. This will need a more thorough analysis that will be done later and published separately.

Reviewer #2 (Remarks to the Author):

The authors have adequately replied to my comments and the new data provided is convincing. I recommend publication with three minor requests that are easy to follow:

1) In my view the new supplementary Figure S16 is the most important data in the whole manuscript because it uses non-patched cells and physiological external solution without K⁺ channel block (presumably leading to physiological resting membrane potentials). This is also acknowledged by the authors in their rebuttal “direct comparison of C2-LC, CoChR, CapChR1 and CapChR2 (Fig. S16), which is in fact more relevant than our screening conditions at high Ca²⁺”. Along with this notion is mandatory to include Fig. S16 in the main manuscript as additional figure (e.g. after Fig. 4). If word count is an issue is hereby requested to the editor to allow this additional figure and legend.

Alternatively, Fig. 4 can be replaced by Fig. S16 because the former it is of less importance showing the same conclusion (but statistics are lacking) as Figure S16 (with statistics) but required patch-clamp induced hyperpolarization, most likely because K⁺ channels are blocked. Please use the same analysis as in Fig 4 (Initial Delta R) for Fig. S16 (currently unclear with “R after illumination”) to be able to compare both figures (and the need for hyperpolarization).

This is an interesting suggestion and we agree that the results merit a spot in the main text. We have accommodated it by merging the previous Figure 4 and S16 into a single Figure in the main text, which is now the new Figure 4 found in the manuscript (see below). We have adjusted the order of the text accordingly. We would prefer to keep the data from Figure 4 in the main text, since it shows the [Ca²⁺]_e influx as Fura-2 response at low physiological Ca²⁺ (2 mM; Figure 4E and F) and relative to other concentrations (Figure 4G and H) for both CapChRs. We have also amended the y-axis labels (deltaR) with corresponding illumination times.

Figure 4: (A-C) Fura-2-AM imaging on CoChR WT (blue), C2-LC (red), CapChR1 (green) and CapChR2 (orange). A) Overview of experimental setup: Fura-2-AM loaded ND7/23 cells with a resting membrane potential were illuminated with 470 nm (~0.08 mW/mm², saturating light intensities) light to allow influx of calcium ions through the expressed ChR in the presence and absence of bath Mg²⁺. B) Mean imaging response of the denoted constructs under the two measured conditions (Mean as colored line and shadows represent S.E.M.). C) Quantified peak responses after 10 s of illumination. Single dots on the

right of the columns represents one cell under those conditions (Bar: Mean \pm S.E.M.; N= 21-32). Two-sided, unpaired Wilcoxon-Mann-Whitney-Test: *P \leq 0.05, **P \leq 0.01, ***P \leq 0.001, ****P \leq 0.0001. (D-H) Voltage-clamped calcium imaging on CapChR1 and 2. D) Experimental design for voltage-clamped measurements on ND7/23 cells. Cells were loaded with membrane-impermeable Fura-2 via the patch pipette (internal buffer: 110 mM [NaCl]_i and divalent cation free, see Methods). A baseline measurement was started for both the 380 nm and 340 nm channels (5 acquisitions), with subsequent 470 nm illumination (100 ms, \sim 0.08 mW/mm², \sim 0.008 mJ/mm² per F_{340/380} ratio acquisition, saturating illumination for both CapChRs) to measure calcium influx through the CapChRs. A membrane voltage of -80 mV was applied at each illumination cycle. E) Calcium imaging response (ratio of 340/380 nm fluorescence; R_{340/380}) for both CapChRs at physiological pH and ion concentrations. F) Fluorescence change (ΔR) after 100 ms of illumination at -80 mV holding potential. G) and H) Fluorescence change (ΔR) for CapChR1 (green) and CapChR2 (orange) after 100 ms of illumination at -80 mV holding potential and at different extracellular calcium and sodium concentrations. Box: Mean \pm S.E.M., Whiskers: 1.5 x S.E.M.). Number of Replicates in G and H: CapChR1/CapChR2: 144 mM [NaCl]_e/0 mM [CaCl₂]_e: n=7/8, 143.8 mM [NaCl]_e/0.1 mM [CaCl₂]_e: n=7/6, 143 mM [NaCl]_e/0.5 mM [CaCl₂]_e: N=7/6, 142 mM [NaCl]_e/1 mM [CaCl₂]_e: n=6/5, 140 mM [NaCl]_e/2 mM [CaCl₂]_e: n=45/29, 134 mM [NaCl]_e/5 mM [CaCl₂]_e: n=7/9, 124 mM [NaCl]_e/10 mM [CaCl₂]_e: n=6/11, 104 mM [NaCl]_e/20 mM [CaCl₂]_e: n=8/5.

For Fig. S16 (and all other figures and legends) please keep abbreviations constant and only use CatCh (as in Fig 1) and not CrChR2-L132C or C2-LC.

We understand the reviewers' suggestion for unitary naming. The naming convention in the paper, that is the usage of CrChR2-L132C and/or C2-LC in figures and in the text instead of CatCh, was chosen to prevent a misunderstanding that CatCh is a useful Ca²⁺-conductor for optogenetic applications. This is rarely so, as shown in this manuscript, in various experiments, and in previous studies (Schneider et al. 2013, PMID: 23823227). The priority over wt CrChR2 at 2 mM Ca²⁺ is very limited and thus, it should not be used when Ca²⁺ influx is intended. The advantage of L132C is the reduced inactivation and larger stationary current as well as the reduced H⁺-conductance. We have however, changed all naming to C2-LC throughout the manuscript to avoid any confusion (except at first mention, where all known names are denoted, including CatCh).

2) The reviewer has commented on the confusing abbreviations and de-novo generation of synonyms and the authors have explained to indicate superior function by indices (example of GCaMP given which by different groups consistently with increasing suffixes). Along this idea it is requested to use CatCh for C2-LC (harmonization as requested still not done), CatCh2 for CapChR1 and CatCh3 for CapChR2. It is just confusing and there is no rationale to introduce the new abbreviation "Calcium permeable ChannelRhodopsins"(CapChR) for a "calcium translocating channelrhodopsin" (CatCh) because "permeable" and "translocating" has essentially the same meaning. By following this request, it will be mandatory for future studies to use CatCh4 and so on for better versions. This would also be best for highlighting the superior function of the new proteins over CatCh.

We do understand the reasoning of the reviewer. However, following the explanation outlined above, we would like to distance our constructs from the naming convention "CatCh", a construct that in the ChR field is known already to be highly Na⁺-permeable (Schneider, Grimm and Hegemann, 2015, PMID: 26098512; Kandori 2020, PMID: 32065378). Furthermore, the name CatCh2 or 3 would imply that we have used the same underlying principle for improvements, which for CatCh is based on cysteine substitutions at H3. As indicated in the original report, L132C faces away from the pore, and is discussed to have an indirect effect on permeability through increased structural flexibility (Kleinlogel et al. 2011, PMID: 21399632). According to our model, this is not the case for CapChR1 and 2. We have introduced carboxylates at the central gate to enhance calcium permeation, according to selectivity filters in natural calcium channels and previous research on ChRs. We believe

that this approach merits distinction from previous efforts, and we would be very grateful if it would be accepted.

3) As previously requested, please provide details on the intracellular solution in sections "Whole-cell patch-clamp recordings in ND7/23 cells" and clearly describe if and how intracellular buffering of Ca^{2+} was adjusted.

Of course, we apologize if this was not included there. We had previously added it to the Figure 4 description ("110 mM $[\text{NaCl}]_i$ and divalent cation free, see Methods"), and was thoroughly listed in the buffers section of the Methods. We have now additionally detailed the intracellular solutions used in the respective methods section "Whole-cell patch-clamp recordings in ND7/23 cells" and "Voltage-clamped Fura-2 calcium imaging in whole-cell configuration". All buffers used and their respective reference Figure can also be found listed in Table 1.

Review signed by Philipp Sasse

Reviewer #3 (Remarks to the Author):

The authors' principal objective was to combine mutations that are known to increase Ca^{2+} permeability in several light-activated ChRs to produce a multi-mutated ChR form with greater permeability. Success was claimed by producing quadruple mutants they named CapChRs with "strongly improved Ca^{2+} permeation". My main concern was that the authors needed to directly compare the Ca^{2+} permeabilities of the CapChRs with the single-mutant ChRs, suggesting in the same conditions as they used to select the best single-mutant ChRs. The authors attempted to make this measurement. However, they report the data acquired were "uninterpretable" (not exceeding noise levels?) under the ionic conditions that were used with success for the single mutants. They changed ionic conditions reducing calcium concentration and found that CapChR1 is not significantly more permeable than the single mutants, but CapChR2 produced ~3-fold higher currents in those conditions. These results improve the manuscript by introducing a direct comparison, but the results are mixed. CapChR2 shows increased permeability, whereas the necessity to change ionic conditions to make the comparison suggests that the single mutants may have larger currents in some conditions and CapChRs in others.

Measurements at 70 mM Ca^{2+} for CapChR1 and 2 induced a saturated Fura-2-fluorescence before blue illumination (470 nm) even began. This is due to the Fura-2-probing UV-pulses causing a high Ca^{2+} influx already, originating from residual CapChR activation in the UV. Thus, these experiments did not provide meaningful insight. Elevated concentrations of extracellular calcium increase the Ca^{2+} -influx too much (Figure 3H and I, Figure 4G and H). The only option to measure Ca^{2+} -influx at high Ca^{2+} in CapChRs would be to strongly buffer Ca^{2+} with BAPTA/EDTA (in addition to Fura-2, which also binds free Ca^{2+}). We have done this in initial testing but not pursued further, because we considered it irrelevant to the applicability of CapChRs at physiological $[\text{Ca}^{2+}]_e$.

A less major concern I had mentioned is that the authors state as a fact that the mutations near the central gate introduce a new Ca^{2+} binding site (Page 21, line 468: "... in conjunction with the new calcium-binding site at the central gate."). I believe their proposed site may be correct, but it is not proven, and it would therefore be more accurate to refer to it as "the calcium-binding site proposed in our model". In the authors' response they disagree seemingly based on the assumption that because the two mutations increase the Ca/Na current ratio, there must be a Ca^{2+} binding site. I suggest reading an article from Park and MacKinnon describing principles favoring the observation

that weakening of an ion binding site may enhance permeability (Park E, MacKinnon R. 2018. Structure of the CLC-1 chloride channel from Homo sapiens. eLife 7:e36629. PMID: 29809153). I do understand that the ITC measurements I had suggested can reasonably be regarded as beyond the scope of this paper.

This is an understandable point and marks an important distinction between models and experimental data. We have now more carefully described the considerations associated with our MD models in the quoted sentence (p21, line 483).

Minor point: The authors refer to 2 papers, one in BioRxiv, reporting discovery of ChRs with high K⁺ selectivity (KCRs). To be complete, there is a third article that reports more KCRs posted in BioRxiv (doi: <https://doi.org/10.1101/2022.09.26.509509>).

Of course, this is added now.